# RULEBREAKERS: Challenging LLMs at the Crossroads between Formal Logic and Human-like Reasoning

Jason Chan [1]    Robert Gaizauskas [1]    Zhixue Zhao [1]

## Abstract

Formal logic enables computers to reason in natural language by representing sentences in symbolic forms and applying rules to derive conclusions. However, in what our study characterizes as "rulebreaker" scenarios, this method can lead to conclusions that are typically not inferred or accepted by humans given their common sense and factual knowledge. Inspired by works in cognitive science, we create RULEBREAKERS, the first dataset for rigorously evaluating the ability of large language models (LLMs) to recognize and respond to rulebreakers (versus non-rulebreakers) in a knowledge-informed and human-like manner. Evaluating seven LLMs, we find that most models achieve mediocre accuracy on RULEBREAKERS and exhibit some tendency to over-rigidly apply logical rules, unlike what is expected from typical human reasoners. Further analysis suggests that this apparent failure is potentially associated with the models' poor utilization of their world knowledge and their attention distribution patterns. Whilst revealing a limitation of current LLMs, our study also provides a timely counterbalance to a growing body of recent works that propose methods relying on formal logic to improve LLMs' general reasoning capabilities, highlighting their risk of further increasing divergence between LLMs and human-like reasoning.

## 1. Introduction

Formal logic has numerous applications in mathematics and computer science (Zach, 2024). In natural language processing (NLP), it enables computers to perform reasoning with natural language sentences by first converting them into

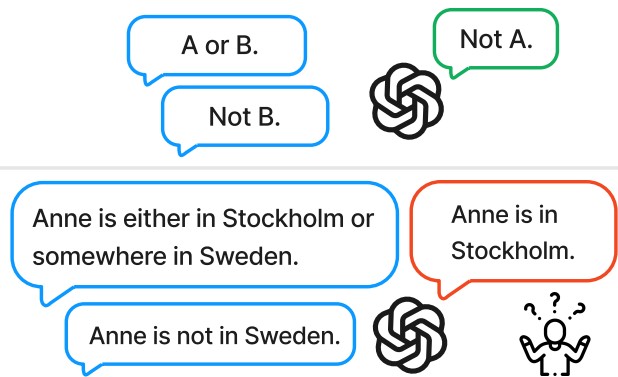

*Figure 1.* An example adapted from Johnson-Laird (1983): rigidly applying logical rules (top) to premises (left) in natural language can result in conclusions (right) that humans would not typically draw or accept, given their common sense and factual knowledge.

symbolic forms (in a process known as semantic parsing) and then applying a set of pre-defined rules to evaluate or derive new conclusions (Kamath & Das, 2019; Saparov & Mitchell, 2022; Olausson et al., 2023, etc.). For example, in a system using propositional logic (Barker-Plummer et al., 2011), sentences are converted into atomic propositions ("*A*", "*B*") optionally connected by one or more of the five connectives ("*not*" (¬), "*and*" (∧), "*or*" (∨), "*if...then*"(→), "*...if and only if...*" (↔)). Given two premises, "*Anne is either in Stockholm or somewhere in Germany*" and "*Anne is not in Germany*", we can convert them symbolically into $A \lor B$ and $\neg B$ respectively. We can then apply the *disjunctive syllogism* rule (for all propositions P and Q, $P \lor Q$, $\neg Q$ ∴ P) to correctly conclude that A i.e. "*Anne is in Stockholm*".

However, as argued in some cognitive science literature (Johnson-Laird, 2010, Ragni & Johnson-Laird, 2018, Khemlani & Johnson-Laird, 2019, etc.), this method of reasoning purely based on logical form and rules is fundamentally different from how humans typically reason with natural language statements. A key problem is that this approach can lead to conclusions which *factually* contradict the premises and which humans typically do *not* draw provided that they recognize this factual contradiction. Drawing on examples adapted from Johnson-Laird (1983) such as Figure 1, suppose we are told that "*Anne is either in Stockholm or somewhere in Sweden*" ($A \lor B$) but we find out in fact that

[1]University of Sheffield, Sheffield, UK. Correspondence to: Zhixue Zhao <zhixue.zhao@sheffield.ac.uk>.

*Proceedings of the 42nd International Conference on Machine Learning*, Vancouver, Canada. PMLR 267, 2025. Copyright 2025 by the author(s).

"*Anne is not in Sweden*" ($\neg B$). Simply applying the *disjunctive syllogism* rule likewise would produce a conclusion that "*Anne is in Stockholm*" ($A$). But most readers, provided that they know Stockholm is located in Sweden, are unlikely to draw this conclusion. This is because, given our common sense and factual knowledge about the world, we recognize that the two premises are inconsistent with each other; and that the proposed conclusion ("*Anne is in Stockholm*") contradicts the second premise ("*Anne is not in Sweden*") in fact and should therefore be rejected. Similarly, suppose we are told that "*if Anne is in Sweden, then she is not in Stockholm*" ($A \rightarrow \neg B$) but we learn in fact that "*Anne is in Stockholm*" ($B$). We would clearly also not just conclude that "*Anne is not in Sweden*" ($\neg A$) by applying the formal rule of *modus tollens* (for all propositions $P$ and $Q$, $P \rightarrow Q$, $\neg Q \therefore \neg P$)[1].

Our study characterizes each of these examples as a "*rulebreaker*": a set of natural language premises and conclusion whereby the conclusion can be derived by applying logical rules to the premises, but is typically not inferred or accepted by humans because, given their common sense and factual knowledge, it is inconsistent with the premises.

In cognitive science, existing studies (Johnson-Laird & Byrne, 2002; Quelhas et al., 2010; Quelhas & Johnson-Laird, 2017) have found that, when human participants are given the premises of rulebreakers similar to the examples above, they indeed typically do not draw what they recognize to be factually contradicting conclusions by rigidly applying a logical rule. By contrast, most participants *do* draw the conclusion in what we call "*non-rulebreaker*" cases (i.e. when the conclusion is not inconsistent with the premises, given common sense and factual knowledge), even though both the rulebreakers and non-rulebreakers have the same surface structure (e.g. "*A or B. Not B. Therefore, A.*"). In NLP, rulebreakers have received little attention, having only been acknowledged in theory (Bos & Markert, 2005; Korman et al., 2018) but not systematically studied.

Our study addresses this gap by creating RULEBREAKERS, a large-scale dataset consisting of *minimally differing* rulebreaker and non-rulebreaker pairs, to rigorously evaluate the reasoning behavior of large language models (LLMs). Specifically, we evaluate the extent to which LLMs can recognize and respond to both rulebreakers and non-rulebreakers in a way that we would expect typical human reasoners to do so, by rejecting conclusions in rulebreakers but accepting those in non-rulebreakers.[2]

Recent studies have suggested that LLMs can perform tasks that require reasoning abilities (see Section 2), but it re-

mains unclear how they do so and whether their reasoning processes are in any way comparable to those of humans. Our study contributes targeted insights to these questions by drawing on cognitive science expertise and assessing LLMs on an under-explored category of reasoning problems.

Moreover, a growing number of recent studies have proposed methods that rely on formal logic to improve LLMs' reasoning capabilities and enable them to perform complex multi-step reasoning. Examples include using logical rules to synthetically generate or augment training data (Wang et al., 2022; Morishita et al., 2024), incorporating logic-based training metrics (Calanzone et al., 2024) or constraints during inference (Weir et al., 2024), and using LLMs to explicitly translate natural language statements into symbolic forms which are then used for the models' subsequent reasoning steps (Xu et al., 2024) or processed by external solvers (Pan et al., 2023; Olausson et al., 2023). Our study provides a timely and important counterbalance to these approaches by drawing attention to their potential pitfalls and trade-off in increasing divergence between LLMs and human-like reasoning.

Our contributions are summarized as follows:

- We develop RULEBREAKERS, a dataset comprising 25,600 instances, along with corresponding evaluation metrics, designed to rigorously assess the reasoning capabilities of LLMs on rulebreakers, an under-explored category of reasoning problems in NLP.[3]

- We are the first to conduct a comprehensive comparison across seven state-of-the-art LLMs, including GPT-4o, on rulebreakers at scale and find that, while LLMs show some capacity to distinguish rulebreakers from non-rulebreakers when reasoning, their overall performance and responses are far from what we expect from typical human reasoners.

- Our subsequent analysis indicates that the models' failure to recognize rulebreakers may be associated with two factors: the models' familiarity with the relevant factual knowledge, and the relative importance they assign to different parts of the premises, opening up promising directions for future investigation.

## 2. Related Work

**Problems with logic as basis of human reasoning**. Existing work in cognitive science, particularly by proponents of the mental model theory[4], have argued against the idea that humans typically reason by identifying the logical form of

---

[1]$B$ is logically equivalent to $\neg(\neg B)$ in propositional logic

[2]Our study refers to this specific reasoning behavior as being "human-like" and "knowledge-informed", interchangeably, on the basis of existing studies in cognitive science as discussed above and further in Appendix A Table 5.

[3]We publicly release RULEBREAKERS at https://github.com/jasonchanly/rulebreakers.

[4]We refer interested readers to Johnson-Laird (1983) for the seminal text on this theory of how humans reason by constructing

natural language premises and then applying logical rules to produce a conclusion (Johnson-Laird, 2010; Khemlani & Johnson-Laird, 2017, etc.). A key reason is that logical rules are defined based on connectives (such as "*if*", "*and*", "*or*") that are assumed to have fixed meanings and properties in formal logic; but in everyday language, the interpretation of connectives varies significantly depending on the semantic content of sentences they connect (Johnson-Laird & Byrne, 2002; Khemlani et al., 2018).[5] Our study is inspired by experimental work that has provided empirical support to these arguments (Johnson-Laird & Byrne, 2002; Quelhas et al., 2010; Quelhas & Johnson-Laird, 2017), which find that human participants generally avoid drawing conclusions that they recognize to be factually contradicting the premises, even where these conclusions might be licenced by a purely formal application of logical rules to the premises.[6] However, while these studies tested human participants on dozens of handcrafted rulebreakers, our study designs rulebreaker templates by which we can systematically generate minimally differing rulebreaker and non-rulebreaker pairs at scale, enabling us to rigorously assess LLMs and control for their world knowledge, bias and sensitivity to prompt variations.

**Reasoning with LLMs**. Reasoning has been characterized as an emergent ability of LLMs (Wei et al., 2022a; Suzgun et al., 2023) and a range of existing work has evaluated the ability of LLMs to perform different types of reasoning, such as logical reasoning (see below), commonsense reasoning (Talmor et al., 2021; Bian et al., 2024) and mathematical reasoning (Meadows et al., 2024; Ahn et al., 2024). Yu et al. (2024), Qiao et al. (2023), Sun et al. (2025), and Mondorf & Plank (2024b) provide further overviews and taxonomies.

Our study is motivated in part by existing work which indicates that LLMs can generally perform logical reasoning using rules in propositional and first-order logic to some degree (Saparov et al., 2023; Han et al., 2024; Xu et al., 2025; Parmar et al., 2024), even though they may struggle with applying particular rules (Parmar et al., 2024), identifying logical fallacies (Wan et al., 2024) and constructing long inference chains (Saparov et al., 2023).

However, our study and dataset has a fundamentally different objective from these prior works, which all assess whether LLMs can identify the underlying logical form of natural language premises, regardless of their semantic content, and apply rules to derive conclusions that are logically

valid purely on the basis of form. In other words, they assume that, given premises in the form of "*if A, then B*" and "*not B*", the conclusion "*not A*" is always correct regardless of what "*A*" and "*B*" symbolize in natural language. Instead, we assess whether LLMs can, like humans (as we argued in Section 1), take semantic content of premises into account when reasoning rather than just rigidly applying formal rules. This means using common sense and factual knowledge to recognize that conclusions in non-rulebreakers are correct (i.e. they are factually consistent with and follow from the premises) whereas those in rulebreakers are not, even though they might appear to have the same logical form such that the same rule seems to apply.

**Comparing human and LLM reasoning**. Our study contributes to the growing interest evidenced by similar recent works in comparing LLMs' reasoning behavior against how humans typically reason according to theories and findings in cognitive science (Suri et al., 2024; Binz & Schulz, 2023; Castello et al., 2024; Mondorf & Plank, 2024a, etc.). Lampinen et al. (2024) found that LLMs exhibit some human-like "content effect" in certain logical reasoning tasks, including the Wason selection task (Wason, 1968) and assessing whether conclusions are valid in syllogisms (arguments in the form of e.g. "*All As are Bs. All Bs are Cs. Therefore, all As are Cs.*"). In other words, models can be mistakenly affected by semantic content when reasoning about problems where only the logical form should be considered: they can be prone to judge conclusions that are true (or plausible) in the real world as logically valid, and conclusions that are false in the real world as invalid, even where the underlying logical structure is the same in both cases. This phenomenon is corroborated by subsequent studies such as ProntoQA (Saparov & He, 2023) and Wu et al. (2024), the latter finding that models tend to perform worse at logical reasoning the more the premises deviate from real-world knowledge. Similarly, Eisape et al. (2024) compares human and LLM behavior in reasoning about syllogisms and found that models mimic some human biases including content effect and figural effect (i.e. judgments being sensitive to the ordering of the premises and the terms within each premise), although larger models exhibit less of these biases and reason in a more logical manner.

Our work differs from the above in that **our focus is not on formal logical reasoning problems that humans might find difficult to answer correctly due to content effect or systematic biases. Rather, the rulebreakers we study are more general reasoning problems in natural language that *do* in fact require models to consider the semantic content of the sentences in order to answer correctly in a knowledge-informed and human-like manner**.

We discuss additional related works in Appendix A.

---

mental models of possibilities, but note that the specific mechanics proposed by the theory itself are not integral to our current work.

[5]In cognitive science, this phenomenon of context-dependent interpretation is also known as the "modulation" of sentential connectives (Johnson-Laird & Byrne, 2002).

[6]Appendix A Table 5 provides further details on these studies. In this sense, rulebreaker cases can also be considered as factual contradictions arising from over-rigid reasoning with formal logic.

*Table 1.* Example **rulebreakers** in the four templates used in our dataset, created by permuting premise-conclusion structures of two logical rules: MT and DS, and two kinds of entity pairs: geographical (country, city) and categorical (type, instance). Note that we create MT rulebreakers in a structure that is different but logically equivalent to the paradigm form "*if P, then Q. Not Q. Therefore, not P.*"

| | Geographical (country, city) | Categorical (type, instance) |
|---|---|---|
| Modus tollens (MT)

*Premises: If P, then not Q.*
*Q.*
*Conclusion: Not P.* | Premises: If Anne is in Sweden,
then she is not in Stockholm.
Anne is in Stockholm.

Conclusion: Anne is not in Sweden. | Premises: If Anne plays some kind of brass instrument,
then she does not play the trumpet.
Anne plays the trumpet.

Conclusion: Anne does not play any kind of brass instrument. |
| Disjunctive syllogism (DS)

*Premises: P or Q.*
*Not Q.*
*Conclusion: P.* | Premises: Anne is in either Stockholm
or somewhere in Sweden.
Anne is not in Sweden.

Conclusion: Anne is in Stockholm. | Premises: Anne plays either the trumpet
or some kind of brass instrument.
Anne does not play any kind of brass instrument.

Conclusion: Anne plays the trumpet. |

## 3. RULEBREAKERS Dataset

We create a dataset consisting of *non-rulebreakers* and *rulebreakers* in equal proportion. We generate rulebreakers using four templates as shown in Table 1. These templates are created in relation to two **logical rules** (modus tollens (MT) and disjunctive syllogism (DS)), with placeholders to be filled with **entity pairs**, **verbs** and **names-pronouns**.

### 3.1. Creating Rulebreakers

**Logical rules** RULEBREAKERS focuses on premise-conclusion structures relating to two rules in propositional logic that have been tested by prior work on human participants (Johnson-Laird & Byrne, 2002; Quelhas et al., 2010; Quelhas & Johnson-Laird, 2017). See Table 1 for examples of rulebreakers with respect to these two rules.[7]

MT: for all propositions $P$ and $Q$, "if $P$ then $Q$" and "not $Q$" entails "not $P$" ($P \rightarrow Q, \neg Q \therefore \neg P$).

DS: for all propositions $P$ and $Q$, "$P$ or $Q$" and "not $Q$" entails "$P$" ($P \lor Q, \neg Q \therefore P$).

**Entity pairs** We include two groups of entity pairs:

- Country and city (geographical pairs): We use a list of current-day countries and their capital cities, obtained by querying WikiData (Vrandečić & Krötzsch, 2014). We filter out capitals that have the same name as their countries (e.g. Singapore) or are located in more than one country, resulting in 183 (country, city) entity pairs.

- Type and instance (categorical pairs): We extract 8 types of entities from ConceptNet (Speer et al., 2017) and manually filter to ensure correctness: birds (20), fish (13), insects (12), brass instruments (9), stringed instruments (16), woodwind instruments (8), martial

arts (8), and racket sports (5). In total, we have 91 (type, instance) pairs, listed in Appendix B.

**Verbs** To fill in the template for creating RULEBREAKERS instances, for country-city pairs, we permute each pair with six manually selected verbs ("*is in*", "*was born in*", "*died in*", "*will be visiting*", "*had studied in*", "*has been to*"). Similarly, we permute each type-instance pair with two manually selected verbs. Specifically, birds, insects, and fish are permuted with "*saw*"/"*caught*", the three kinds of musical instruments with "*plays*"/"*owns*", and martial arts and racket sports with "*is good at*"/"*is competing in*".

**Names-pronouns** We further permute each of these entity pair + verb combinations with five given names and corresponding pronouns, randomly sampled without replacement from a list of common first names worldwide[8].

### 3.2. Creating Non-rulebreakers

We create a non-rulebreaker counterpart for each rulebreaker by randomly replacing the country or type used in the premises and conclusion with a different country or type, respectively. For example, we create the following non-rulebreaker by replacing "France" with "Germany":

> Premises: If Anne is in ~~France~~Germany, then she is not in Paris. Anne is in Paris.
>
> Conclusion: Anne is not in ~~France~~Germany.

In this case, the conclusion, which can be derived by applying MT, follows from and is consistent with the premises, given common sense and factual knowledge about the world.

### 3.3. Dataset Summary

In total, RULEBREAKERS consists of 12,800 rulebreakers, created according to Section 3.1, with the breakdown

---

[7]Appendix A Table 5 shows how the templates used in creating our RULEBREAKERS dataset correspond to specific examples used in these prior experimental work.

[8]https://github.com/sigpwned/popular-names-by-country-dataset/

shown in Figure 2. We then create 12,800 non-rulebreakers per Section 3.2, bringing the combined number of rulebreakers and non-rulebreakers in RULEBREAKERS to 25,600.[9]

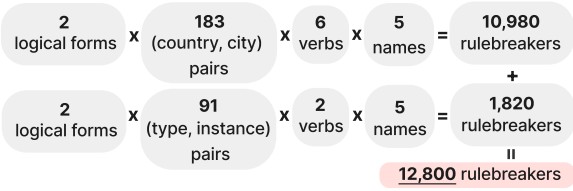

*Figure 2.* Statistical summary of rulebreakers in our dataset.

### 3.4. Evaluation

We measure a model's ability to distinguish and respond to rulebreakers and non-rulebreakers in a human-like manner in terms of (a) accuracy; and (b) model confidence.

In formal notation, let $D$ represent the set of prompts used in our experiment. Specifically, $D$ comprises all instances in RULEBREAKERS, each permuted across 10 distinct phrasing variations, as detailed in Section 4.2. We define $D^R$ and $D^N$ as the sets of rulebreakers and non-rulebreakers respectively within $D$, and $D^{paired}$ as the subset of $D^R \times D^N$ such that for all $(x^R, x^N) \in D^{paired}$, $x^N$ is the non-rulebreaker counterpart to $x^R$, as described in 3.2.

**Accuracy**   We first define **paired accuracy** as the proportion of all rulebreaker and non-rulebreaker prompt *pairs* $(x^R, x^N)$ in $D^{paired}$ where both parts of the pair are correctly answered by a model. Pairs are defined such that the rulebreaker and non-rulebreaker counterparts differ only by country or type name.[10] Thus, a pair is answered correctly if and only if the model outputs "no" for the rulebreaker (i.e. rejecting the conclusion) and "yes" for the non-rulebreaker prompt (accepting the conclusion), or "false" and "true" respectively where we specify those as answer options per Section 4.2. Formally, paired accuracy ($\tau$) is defined as:

$$\tau = \frac{1}{|D^{paired}|} \sum_{(x^R, x^N) \in D^{paired}} [\![T(x^R) \wedge T(x^N)]\!] \quad (1)$$

where $T(x)$ is true if and only if the model outputs the correct answer to the prompt $x$. This paired accuracy metric is designed to reflect the core objective of our evaluation and control for any inherent biases in model responses. Specifically, if a model completely fails to distinguish between rulebreakers and non-rulebreakers, e.g., if it outputs "yes"/"true" for every instance in $D$ because of a strong inherent bias for

positive over negative answers, it would yield a score of 0. Accordingly, the baseline for random guessing is 0.25.

We also measure the rulebreaker-specific and non-rulebreaker-specific accuracies ($\tau_{D^R}$ and $\tau_{D^N}$). These are defined as the proportion of correctly answered rulebreaker and non-rulebreaker instances, respectively. Formally:

$$\tau_{D^R} = \frac{1}{|D^R|} \sum_{x \in D^R} [\![T(x)]\!] \quad (2)$$

$$\tau_{D^N} = \frac{1}{|D^N|} \sum_{x \in D^N} [\![T(x)]\!] \quad (3)$$

**Model confidence**   Inspired by Lampinen et al. (2024), to gain a more fine-grained understanding of a model's behavior, we evaluate its "confidence" in responding to both rulebreakers and non-rulebreakers. Consider a scenario where a model consistently outputs "yes" to both non-rulebreaker and rulebreaker prompts but its confidence levels in its "yes" answers to non-rulebreakers are significantly higher compared to rulebreakers. In this case, despite achieving a paired accuracy of 0, the discrepancy in model confidence may still indicate some underdeveloped latent ability to differentiate between rulebreakers and non-rulebreakers.

For each prompt $x$ in $D$, we compute the model's confidence in either the word "yes" or "true" (depending on the answer options specified in $x$, as Section 4.2 will explain) by extracting the output probabilities of tokens corresponding to that word (e.g. "Yes"/"YES"/"yes")[11]. We refer to this value as the **model's confidence in a positive answer** to a given prompt $x$: $p^+(x)$. Formally, where the answer options specified in $x$ are "yes"/"no":

$$p^+(x) = \sum_{\hat{y} \in \{``Yes", ``yes", ``YES"\}} p(\hat{y}|x) \quad (4)$$

where $p(\hat{y}|x)$ refers to the output probability the model assigns to the token $y$ given a prompt $x$. Likewise, where the specified answer options are "true"/"false", $p^+(x)$ is defined by the summed probabilities of "True"/"true"/"TRUE".

We aim to evaluate whether the model, on average, exhibits higher confidence when outputting "yes"/"true" for non-rulebreakers (a correct response) compared to when it outputs "yes"/"true" for rulebreakers (an incorrect response). To do this, we partition $D$ into $D^{yes}$ and $D^{no}$, which represent the subsets of prompts in $D$ where the model gives positive ("yes"/"true") and negative ("no"/"false") answers, respectively. We then compute and compare $\Pi^+_{D^N}$ and $\Pi^+_{D^R}$:

$$\Pi^+_{D^N} = \frac{\sum_{x \in (D^N \cap D^{yes})} p^+(x)}{|D^N \cap D^{yes}|} \quad (5)$$

---

[9]Appendix C lists the licenses of the various underlying datasets used in creating RULEBREAKERS as described above.

[10]For example, "*If Anne is in France, then she is not in Paris...*" (rulebreaker) and "*If Anne is in Germany, then she is not in Paris...*" (non-rulebreaker) would constitute a pair.

[11]This is to account for the issue of surface form competition as highlighted in Holtzman et al. (2021).

$$\Pi^+_{D^R} = \frac{\sum_{x \in (D^R \cap D^{yes})} p^+(x)}{|D^R \cap D^{yes}|} \qquad (6)$$

Ideally, models should have a $\Pi^+_{D^N}$ as close to 1, and $\Pi^+_{D^R}$ as close to 0 as possible.[12] We test for significance using Welch's *t*-test (Welch, 1947), given that the sample sizes are likely to be unequal: a model might output "yes"/"true" to more non-rulebreakers than to rulebreakers, i.e. $|D^N \cap D^{yes}| > |D^R \cap D^{yes}|$, or vice versa.

## 4. Experiment

### 4.1. Models

We limit our model selection to those that are (a) medium-sized, due to computational constraints; (b) instruction-fine-tuned, so that their outputs are more likely to follow instructions in our prompts; and (c) open-sourced, to ensure access to their internal weights and hidden states for further analysis. Based on these criteria, we select six models from the top of the Open LLM Leaderboard on Hugging Face at the time of writing[13]: Microsoft-Phi-3-mini-128k-Instruct and Microsoft-Phi-3-medium-128k-Instruct (Abdin et al., 2024), Meta-Llama-3-8B-Instruct and Meta-Llama-3-70B-Instruct (Dubey et al., 2024), Mistral-7B-Instruct-v0.3 (Jiang et al., 2023), and Gemma-2-27b-it (Gemma Team et al., 2024).[14] For comparison, we also conduct a limited evaluation of a popular close-sourced model, GPT-4o (gpt-4o-2024-11-20) (OpenAI, 2024) on RULEBREAKERS, given that we do not have access to its full output probability distribution and internal states.

To ensure the model's failure to recognize rulebreakers is not due to a lack of knowledge about the entities involved, we filter out those entity pairs which the model lacks relational knowledge of. For example, an LLM needs to know that the city is situated within its corresponding country to recognize when the conclusion is factually inconsistent with the premises. Therefore, we prompt each of the seven LLMs with "*Complete the sentence: [city] is in*" for each (country, city) pair and "*Complete the sentence: [instance] is a type of*" for each (type, instance) pair. Using pattern-matching, we identified 20 city names for which at least one model did not produce the expected answer (e.g., some LLMs interpret "Georgetown" to refer to the university rather than the capital city of Guyana). We exclude the subset of prompts in $D$ containing any of these 20 city names. We perform a similar check for (type, instance) pairs (see Appendix E) but did not identify any instances that need filtering.

### 4.2. Prompting Setup

To account for LLMs' sensitivity to prompt phrasing (Zhao et al., 2021, Lu et al., 2022, Li et al., 2024), we present each instance in RULEBREAKERS in 10 different variations, by permuting five basic phrasings:

1. Does the Conclusion *follow from* the Premises?
2. Do the Premises *entail* the Conclusion?
3. Can the Conclusion *be inferred from* the Premises?
4. Can the Conclusion *be deduced from* the Premises?
5. Do the Premises *support* the Conclusion?

with two answer-option pairs to follow the main question ("*Answer Yes or No only.*" or "*Answer True or False only.*"[15]). Section 5 reports the averaged results across these 10 phrasing variations, with a breakdown shown in Appendix F.

We apply the chat template specific to each model, without including any system prompts. Example prompts used are in Appendix G. Using the default configurations, we perform one forward pass on each prompt to extract the most probable output token and the output probability distribution.

For generalizability, we also conduct further experiments with variations of this setup. These include adding instructions and further modifying prompt/question phrasings (see Appendix H); and also prompting LLMs to generate a conclusion from the premises themselves instead of simply choosing whether or not to accept a given conclusion (see Appendix I).

### 4.3. Implementation

For all open-sourced LLMs, we use the model version hosted on Hugging Face (Wolf et al., 2020). All experiments are performed on NVIDIA A100 GPUs, with further details in Appendix J. We access GPT-4o via API.[16]

## 5. Results

### 5.1. Accuracy

As shown in Figure 3, while a majority of LLMs achieve an overall paired accuracy above the baseline (0.25), only Meta-Llama-3-8B-Instruct achieves a paired accuracy greater than 0.6, indicating substantial room for improvement for all LLMs' reasoning capabilities. Notably, GPT-4o underperforms Phi-3-mini-128k-Instruct in paired accuracy and even scores the lowest 0.0022 on the MT categorical-entity pairs subset compared to all other models.

---

[12]Where a perfect model answers all rulebreakers correctly with "no"/"false", $|D^R \cap D^{yes}| = 0$ and $\Pi^+_{D^R}$ is undefined.

[13]https://huggingface.co/spaces/open-llm-leaderboard/open_llm_leaderboard

[14]Further details on model size are provided in Appendix D.

[15]We rephrase true/false questions into this form: "*Is it True or False that [the Conclusion follows from the Premises]?*"

[16]Unless specified otherwise, experiments with GPT-4o were conducted in January 2025.

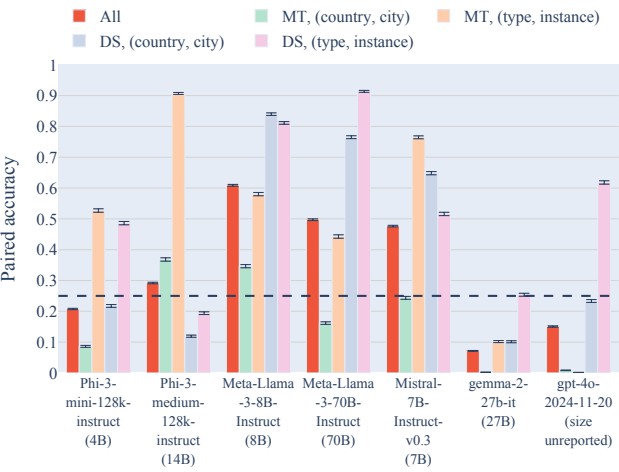

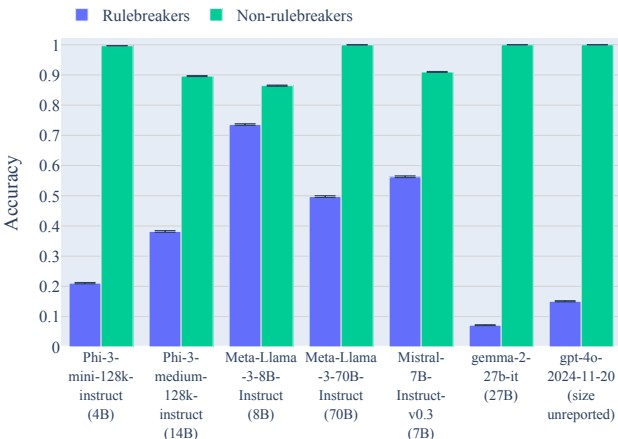

*Figure 3.* Paired accuracy achieved by each LLM (95% CI). The overall paired accuracy achieved over all prompts is shown in red, while the paired accuracy with respect to different subsets of prompts are shown in other colours. Subsets of prompts are defined with respect to the logical rule (MT or DS) and kind of entity pair used in each prompt, as set out earlier in Table 1.

Figure 4 shows that the poor-to-mediocre paired accuracy results can be attributed primarily to the models' poor performance with respect to rulebreakers. On the one hand, all models perform well on non-rulebreakers, correctly predicting that the conclusion follows from the premises in these cases. Gemma-2-27b-it and GPT-4o, for example, even achieve perfect scores. On the other hand, all models perform worse on rulebreaker prompts to various degrees. In particular, three models that perform particularly well on non-rulebreakers, Phi-3-mini-128k-Instruct, Gemma-2-27b-it and GPT-4o, also perform the worst on rulebreakers. This suggests that these models may indeed be exhibiting some degree of "over-generalizing" the two logical rules in our study, MT and DS, rigidly applying these rules unlike what is typically expected of knowledge-informed human reasoners. In other words, **while these models exhibit reasoning behavior that appears highly logical and robust against variations in semantic content, this same trait also appears to undermine their ability to recognize when semantic content should in fact be taken into account when reasoning.**

In light of these accuracy results, we also conduct further experiments to investigate how models that have been "enhanced" using various logic-related methods (e.g. fine-tuning on a logic dataset, integrating the model with an external symbolic solver) would perform on RULEBREAKERS (see Appendix K).

Furthermore, we see a high degree of variability in terms of each model's performance on different prompt subsets. Generally, we observe that models achieve higher paired accuracy on the DS subsets compared to MT, and achieve higher paired accuracy on prompts containing categorical

*Figure 4.* Accuracy each LLM achieves on all rulebreakers ($\tau_{D^R}$) and non-rulebreakers ($\tau_{D^N}$) in $D$ (95% CI).

rather than geographical entity pairs. As we will further explore in Section 6, we conjecture that a model's ability to recognize rulebreakers may be influenced by its "familiarity" with entities mentioned in the prompts (this might be due to e.g. specific city and country names having appeared less frequently in the models' pre-training data than generic nouns, e.g. "wasp"). Nonetheless, the fact that these trends are far from universal suggests that model performance on RULEBREAKERS is driven by multiple competing factors, which highlights a challenge for further analysis.

**5.2. Model Confidence**

As shown in Table 2, the value of $\Pi_{D^N}^+$ is greater than $\Pi_{D^R}^+$ for all models ($p < 0.0001$). That is, all models are on average more confident in their positive answer ("yes"/"true") to non-rulebreakers (which is the correct answer) than in their positive answer to rulebreakers (which is the incorrect answer). This suggests that all models in our study possess a latent ability to distinguish rulebreakers from non-rulebreakers, even where this is not evident from evaluating models only based on their final output tokens. GPT-4o is excluded from this analysis since our methodology (see Section 3.4) requires access to the probability of specific tokens regardless of where they rank in the model's predictions.

**6. Analysis**

This section investigates potential factors that contribute to suboptimal performance of the LLMs. That is, we are interested in why distinguishing between rulebreakers and non-rulebreakers poses such a challenge to these models.

To correctly distinguish rulebreakers and non-rulebreakers, intuitively, a model needs to meet two necessary but not sufficient conditions: (1) accurate and confident knowledge of entities mentioned in the prompts, such as correctly associating a city with its country; and (2) sufficient attention to

*Table 2.* For each model, we compare the model's mean confidence (%) in a positive answer ("yes"/"true") between (a) the subset of rulebreaker prompts to which it outputs a positive answer ($\Pi^+_{D^R}$); and (b) the subset of non-rulebreaker prompts to which it outputs a positive answer ($\Pi^+_{D^N}$). Std dev in brackets.

| MODEL | $\Pi^+_{D^R}$ | $\Pi^+_{D^N}$ |
|---|---|---|
| PHI-3-MINI-128K-INSTRUCT | 92.055 (8.857) | **96.224** (2.738) |
| PHI-3-MEDIUM-128K-INSTRUCT | 93.955 (11.699) | **97.158** (8.763) |
| META-LLAMA-3-8B-INSTRUCT | 77.202 (15.011) | **90.457** (12.095) |
| META-LLAMA-3-70B-INSTRUCT | 96.336 (9.812) | **99.950** (0.366) |
| MISTRAL-7B-INSTRUCT-V0.3 | 92.553 (12.863) | **98.106** (6.925) |
| GEMMA-2-27B-IT | 97.992 (6.074) | **99.996** (0.057) |

the "factual" part of the premises (e.g. "*Anne is in Paris*") that implicitly contradicts the conclusion (e.g. "*Anne is not in France*") in case of rulebreakers.

Accordingly, we propose two hypotheses: *(1) failure to recognize rulebreakers is associated with the model's lack of confidence in its knowledge of the entities involved*[17]; and *(2) this failure is associated with insufficient attention given to the factual premise.* The intuition is that when a model is not sufficiently "familiar" with the entities mentioned in the prompt (i.e. sufficiently confident in its knowledge about the entities), then it will tend to treat these entities symbolically and focus more on the surface form rather than the semantic content of the prompt, thereby overlooking the significance of the factual premise which implicitly contradicts the conclusion in case of rulebreakers.[18]

*Is failure associated with the model's lack of confidence in its knowledge?*

To answer this, we prompt each LLM with the instruction: "*Complete the sentence: [city] is in*" for each city in RULEBREAKERS that all LLMs have answered correctly as part of the filtering process in Section 4.1. We extract the output probability of the first token of the country name, which we refer to as the model's "familiarity" with the particular (country, city). We compare each LLM's mean "familiarity" between prompt pairs in $D^{paired}$ the model `recognizes` against pairs the model `fails` , as shown in Table 3.[19]

---

[17]This is inspired in part by Neeman et al. (2023) investigating how models resolve conflict between in-context information and parametric knowledge.

[18]Providing support to this intuition, Johnson-Laird & Byrne (2002) found that human participants are also more likely *not* to draw factually inconsistent conclusions when given rulebreakers with familiar entities as opposed to unfamiliar ones.

[19]As per Section 3.1, each [city] is mentioned in 600 (rule-

*Table 3.* For each model, we extract its familiarity with the [city] entity mentioned in each prompt pair in $D^{paired}$, and compute the mean (and std dev) corresponding to those prompt pairs the model answers incorrectly, against those it answers correctly.

| | CORRECT PAIRS | INCORRECT PAIRS |
|---|---|---|
| PHI-3-MINI-128K-INSTRUCT | **0.906 (0.128)** | 0.872 (0.172) |
| PHI-3-MEDIUM-128K-INSTRUCT | **0.922 (0.089)** | 0.912 (0.104) |
| LLAMA-3-8B-INSTRUCT | 0.837 (0.187) | **0.844 (0.189)** |
| LLAMA-3-70B-INSTRUCT | **0.959 (0.074)** | 0.949 (0.081) |
| MISTRAL-7B-INSTRUCT-V0.3 | **0.977 (0.072)** | 0.969 (0.085) |
| GEMMA-2-27B-IT | **0.997 (0.032)** | 0.992 (0.048) |
| GPT-4O | 0.909 (0.134) | **0.914 (0.136)** |

If our hypothesis (1) is correct, we would expect each model to have a higher "familiarity" with respect to prompt pairs in its "recognized" group, i.e. those that the model has answered correctly, as compared to its "failed" group. As shown in Table 3, this is the case for most LLMs, except for Meta-Llama-3-8B-Instruct and GPT-4o (Welch's *t*-test $p < 0.001$). While Meta-Llama-3-8B-Instruct's deviation from the predominant pattern might be explained by its relatively high sensitivity to prompt variations (as further discussed in Appendix F), we recognize however that this is not the case for GPT-4o, which is one of the models least sensitive to prompt variations.

As an additional observation, when comparing overall familiarity levels across models, we see that simply being familiar with entities does not guarantee that a model will perform well on RULEBREAKERS. A particular example is Gemma-2-27b-it, which is on average more familiar with entities in our dataset compared to all other models (in terms of both its "failed" and "recognized" groups) but is among the worst-performing models in terms of paired accuracy. This is also despite the perfect accuracy it has achieved on the non-rulebreaker subset of the prompts (see Section 5.1). When combined, the two observations aptly characterize the blind spot that our study has highlighted with LLMs' reasoning: **a model may excel at both retrieving factual knowledge and applying rules in formal logic, yet fail to recognize where rigidly applying these logical rules would produce conclusions that factually contradict the premises.**

We conduct a qualitative analysis on the most common failure instances across models, observing further evidence

---

breaker, non-rulebreaker) pairs (permuting 6 verbs, 5 first names, 2 logical forms and 10 prompt phrasing variations).

*Table 4.* Mean normalized score ratios (with sd) of each LLM. Paired accuracy ($\tau$) from Section 5.1 is displayed for analysis. The greater the ratio, the more the model attends to the factual context.

| | SCORE RATIO (INPUT $\times$ GRADIENT) | SCORE RATIO (ATTENTION) | $\tau$ |
|---|---|---|---|
| PHI-3-MINI-128K-INSTRUCT | 0.738 (0.204) | 1.037 (0.124) | 0.208 |
| PHI-3-MEDIUM-128K-INSTRUCT | 0.657 (0.156) | 1.065 (0.534) | 0.292 |
| META-LLAMA-3-8B-INSTRUCT | 0.814 (0.149) | **1.829 (0.200)** | **0.609** |
| META-LLAMA-3-70B-INSTRUCT | **0.827 (0.201)** | 1.618 (0.182) | 0.497 |
| MISTRAL-7B-INSTRUCT-v0.3 | 0.740 (0.163) | 1.397 (0.145) | 0.476 |
| GEMMA-2-27B-IT | 0.787 (0.193) | 1.558 (0.274) | 0.071 |

of our hypothesis. Detailed discussion and examples are provided in Appendix L.

*Is failure associated with the model overlooking the factual information in the second premise?*

To answer this, we apply feature attribution methods (*input $\times$ gradient* (Simonyan et al., 2014) and *attention* (Bahdanau et al., 2015))[20] to assign an importance score to each token in a prompt. We then sum the importance score of tokens within the second premise, and the first premise, separately. To ensure a fair comparison between the two, we normalize the importance score by the number of tokens aggregated, since the second premise (e.g. "*Anne is in Paris*") is always shorter than the first premise (e.g. "*If Anne is in France, then she is not in Paris.*"). We then compute a "*score ratio*" by dividing the normalized importance score of the second premise by the normalized importance score of the first. Therefore, the lower the ratio, the less importance is assigned to the second premise (factual context) by the model. For each model, we compute a score ratio averaged across all prompts. GPT-4o is excluded since we do not have access to the model's internal states or gradient information.

Our second hypothesis is that low paired accuracy is associated with insufficient attention a model assigns to the factual information in the second premise. As shown in Table 4, this appears to be the case when comparing Phi-3-medium-128k-instruct, Mistral-7B-Instruct-v0.3 and Meta-Llama-3-70B-Instruct: the greater the $\tau$, the higher the score ratio, indicating the greater attention to the second premise. The two models with the highest $\tau$, Meta-Llama-3-8B-Instruct and Meta-Llama-3-70B-Instruct, also have the two highest score ratios. However, Gemma-2-27b-it notably has the third highest score ratio (whether computed by input $\times$ gradient or attention) but exhibits the lowest $\tau$ by a substantial margin.[21] These findings suggest that the relation between failure and attention distribution is less clear-cut than expected. Similar to the "familiarity" factor investigated in

our hypothesis (1), the general though not universal pattern observed in attention distribution suggests that it is one of multiple competing factors influencing model performance on RULEBREAKERS, warranting further investigation in future work.

We conduct a further analysis experiment with respect to neuron activations and discuss our findings in Appendix N.

## 7. Conclusion

We present RULEBREAKERS, a large-scale dataset for evaluating LLMs' ability to reason with natural language in a knowledge-informed and human-like manner by utilizing common sense and factual knowledge rather than rigidly applying rules prescribed by formal logic. In particular, our study of seven LLMs reveals a significant blind spot in their ability to recognize and reject conclusions that can be trivially derived using certain rules in propositional logic but are nonetheless factually inconsistent with the premises. Whilst robust logical reasoning may be desirable in certain applications, our findings highlight the gap between this approach to reasoning and how humans typically reason with natural language in a knowledge-informed manner, drawing attention to an important trade-off in relying on logic-based methods and constraints to improve LLMs' general reasoning capabilities.

## Acknowledgement

This work was supported by the UKRI AI Centre for Doctoral Training in Speech and Language Technologies (SLT) and their Applications funded by UK Research and Innovation [grant number EP/S023062/1]. For the purpose of open access, the author has applied a Creative Commons Attribution (CC BY) licence to any Author Accepted Manuscript version arising. We acknowledge IT Services at The University of Sheffield for the provision of services for High Performance Computing.

## Impact Statement

This work advances our understanding of language models' reasoning capabilities, revealing critical implications for AI safety and system robustness. Our findings directly inform the development of more reliable AI systems through concrete metrics and benchmarks. These contributions are crucial as AI increasingly drives high-stake decisions where alignment with human reasoning processes and judgments is essential.

---

[20]*Input $\times$ gradient* has been shown to be reasonably faithful across models and datasets (Zhao et al., 2022, Zhao & Aletras, 2023). To compare, we also compute a separate score ratio using attention weights (Bahdanau et al., 2015). We implement both methods with *inseq* v0.7.0 (Sarti et al., 2023).

[21]We also analyzed unnormalized score ratios (see Appendix M) but did not find any substantive deviation from these observations.

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

## A. Additional Related Work

**Further comparisons between RULEBREAKERS and existing datasets in logical reasoning**. On the surface, RULEBREAKERS seems similar to logical reasoning datasets introduced in Parmar et al. (2024) and Holliday et al. (2024) in that they also focus on single-step inference instead of long reasoning chains. It also seems similar to other logical reasoning datasets such as LogicNLI (Tian et al., 2021) and ProntoQA (Saparov & He, 2023) in that sentences within the dataset are also syntactically straightforward and easy to represent in symbolic forms in formal logic.

However, as Section 2 has explained in detail, RULEBREAKERS has a fundamentally different nature and objectives from these logical reasoning datasets. It assesses LLMs' ability to utilize common sense and factual knowledge *about the real world* when reasoning, not just how rigidly they can apply logical rules. For this reason, our examples also do not introduce any fictional ontology, nonsensical entities or deliberately false premises (e.g. "*All mammals are cats*")[22], unlike some logical reasoning datasets such as ProntoQA (Saparov & He, 2023) or Wu et al. (2024).

**Failure modes of LLMs.** LLMs have exhibited surprisingly simple failure modes, such as being sensitive to premise order (Chen et al., 2024) and unable to infer that "*B is A*" upon learning that "*A is B*" (Berglund et al., 2024). By examining LLMs' reasoning behavior in the context of rulebreakers, our work also sheds light on another potential limitation of these models: their propensity to "over-generalize" certain logical rules learned from patterns in their training data, potentially applying them incorrectly or too rigidly in reasoning tasks.

Concurrent to our work, Holliday et al. (2024) evaluated LLMs on reasoning problems in conditional and modal logic, including some "*controversial*" cases which we would also consider to be rulebreakers under our definition. However, there are key differences between their contributions and ours. Firstly from a theoretical perspective, while they consider these cases as specific anomalies that are subject to competing theories in logic and philosophy, we draw on expertise in cognitive science to characterize these as part of a larger category of examples that demonstrate the general gap between formal logic and human-like reasoning. Accordingly, we draw attention to scenarios where even basic rules in propositional logic which they do not consider to be "controversial" (e.g. disjunctive syllogism) can produce problematic inferences when applied rigidly in reasoning with natural language. Moreover, while Holliday et al. (2024) gathered data instances for each logical rule by handcrafting one question and then generating dozens more in the same form with the help of LLMs, our study's novel design and use of rulebreaker *templates* enables us to generate instances *at scale* that are carefully controlled – this allows for more in-depth and rigorous analysis into LLMs' reasoning behavior with respect to these problems.

Table 5 provides an overview of the relevant studies conducted in cognitive science relating to what we have termed as "rulebreakers". We show how templates introduced by our RULEBREAKERS are abstracted from specific examples that were manually written and tested on human participants in these studies.[23]

## B. Categorical Entity Pairs in the RULEBREAKERS Dataset

| Type | Instances |
|---|---|
| bird | finch, robin, ostrich, parakeet, swan, pheasant, crow, falcon, goose, nightingale, pelican, hawk, puffin, peacock, flamingo, quail, gull, toucan, partridge, kingfisher |
| fish | trout, crappie, pike, tuna, catfish, flounder, mullet, salmon, haddock, hake, eel, goldfish, cod |
| insect | fly, butterfly, ant, earwig, beetle, flea, firefly, stonefly, termite, wasp, bee, mayfly |
| stringed instrument | banjo, sitar, koto, samisen, zither, guitar, dulcimer, psaltery, violin, gusli, lute, harp, balalaika, cello, mandolin, viola |
| brass instrument | tuba, trumpet, euphonium, trombone, sousaphone, French horn, cornet, bugle, flugelhorn |
| woodwind instrument | clarinet, oboe, flute, piccolo, bassoon, bagpipe, shakuhachi, English horn |
| martial art | tai chi, karate, aikido, kung fu, judo, tae kwon do, jiu jitsu, muay thai |
| racket sport | tennis, badminton, table tennis, squash, racquetball |

---

[22]For example, "*If Anne was born in Sweden, then she was not born in Stockholm*" does not imply that Stockholm is not in Sweden - rather, it only implies that Anne might be born somewhere else in Sweden, or in another country.

[23]We also carried out an informal inspection/annotation on a subset of samples generated with our RULEBREAKERS templates, to validate that they mirror the semantic patterns of manually written examples used in these prior studies.

*Table 5.* Key studies on rulebreakers in cognitive science and their connection to various templates we have introduced in our RULE-BREAKERS dataset.

| Study | Relevant examples from their dataset | Our corresponding rulebreaker template(s) | No. of human participants included in study | Relevant finding |
|---|---|---|---|---|
| Quelhas et al., 2010 | "*If Ana drinks a juice, then she doesn't drink an orange juice*"; "*If Manuel plays a game, then he doesn't play football*" | MT (type, instance) | 28 | Participants generally avoid conclusions that factually contradict the premises, even where they can be derived by applying a logical rule (e.g. modus tollens). Instead, they conclude that "nothing follows" from the premises. |
| Quelhas & Johnson-Laird, 2017 | "*Andre is in Lisbon or he is in Portugal*"; "*Luis is cooking bass or he is cooking fish*"; "*Luis is eating chicken or he is eating meat*" | DS (country, city), DS (type, instance) | 80 | As above, participants generally avoid conclusions that factually contradict the premises, even where they can be derived by applying a logical rule (disjunctive syllogism). |
| Johnson-Laird & Byrne, 2002 | "*If Bill is in Brazil then he is not in Rio de Janeiro*"; "*If Ann is in the Hotel LaBlanc then she is not in the Champagne Suite*" | MT (country, city) | 41 | When participants are familiar with the entities mentioned in the premises, they are more likely to recognize and avoid factually contradicting conclusions, as compared to when they are unfamiliar with the entities. |

## C. Licenses of Datasets Used in Creating RULEBREAKERS

We list in Table 6 existing datasets that we used in creating RULEBREAKERS, and their respective licenses.

*Table 6.* Datasets used in creating RULEBREAKERS and their corresponding licenses

| Dataset | Usage in RULEBREAKERS | License |
|---|---|---|
| WikiData | Source of geographical entity pairs (list of current-day countries and capital cities) | CC0-1.0 Universal |
| ConceptNet | Source of categorical entity pairs | CC BY-SA 4.0 |
| Popular Names by Country Dataset | List of most common first names and corresponding pronouns around the world | CC0-1.0 Universal |

## D. Model Sizes

Table 7 lists the six open-sourced LLMs included in our study, and their respective number of parameters.

## E. Checking Models' Knowledge with Respect to Categorical Entity Pairs

Compared to the check for geographical (country, city) pairs, the check for categorical (type, instance) pairs elicited more varied responses from the models. While some of these responses did not precisely match the ground truth (i.e. the specific type that an entity belongs to according to our dataset), they are technically correct and demonstrate that the model has the relevant knowledge. For example, when prompted with "*Complete the sentence: goose is a type of*", some models responded not with "bird" (which is the ground truth type in our dataset) but with "waterfowl", which is nonetheless correct,

*Table 7.* Number of parameters of each open-sourced LLM used in our study.

| MODEL NAME | NO. OF PARAMETERS |
| --- | --- |
| PHI-3-MINI-128K-INSTRUCT | $3.8 \times 10^9$ |
| PHI-3-MEDIUM-128K-INSTRUCT | $1.4 \times 10^{10}$ |
| META-LLAMA-3-8B-INSTRUCT | $8.0 \times 10^9$ |
| META-LLAMA-3-70B-INSTRUCT | $7.1 \times 10^{10}$ |
| MISTRAL-7B-INSTRUCT-V0.3 | $7.3 \times 10^9$ |
| GEMMA-2-27B-IT | $2.7 \times 10^{10}$ |

since waterfowl is a subspecies of birds.

Consequently, we perform an additional manual check and, on this basis, did not exclude any type-instance pairs as part of this filtering process.

## F. Results Breakdown by Prompt Phrasing

As shown in Figure 5, the seven LLMs exhibit varying degrees of sensitivity to input prompt phrasings used in our study. We find that Meta-Llama-3-8B-Instruct is the most sensitive in this regard, with paired accuracy varying by over 0.4. On the other end of the spectrum, we find that Phi-3-medium-128k-instruct, GPT-4o and Gemma-2-27b-it are the most robust to the prompt phrasing variations included in our study, although the latter two are also the worst performing models in terms of paired accuracy.

An interesting observation is that we did not identify any particular correlation between model performance and the different verbs used in the phrasings. As shown in Figure 6, verbs that elicit the best (or worst) accuracy on rulebreakers and non-rulebreakers differ among the seven LLMs. This contrasts with an intuitive expectation that e.g. the word "support" might be eliciting softer reasoning by a model or that, conversely, the words "entail" and "deduced", which are more often used in the context of logical reasoning, might be motivating the model to adhere more strictly to logical rules when reasoning (and therefore perform worse on rulebreakers). We leave a more detailed investigation of this phenomenon to future work.

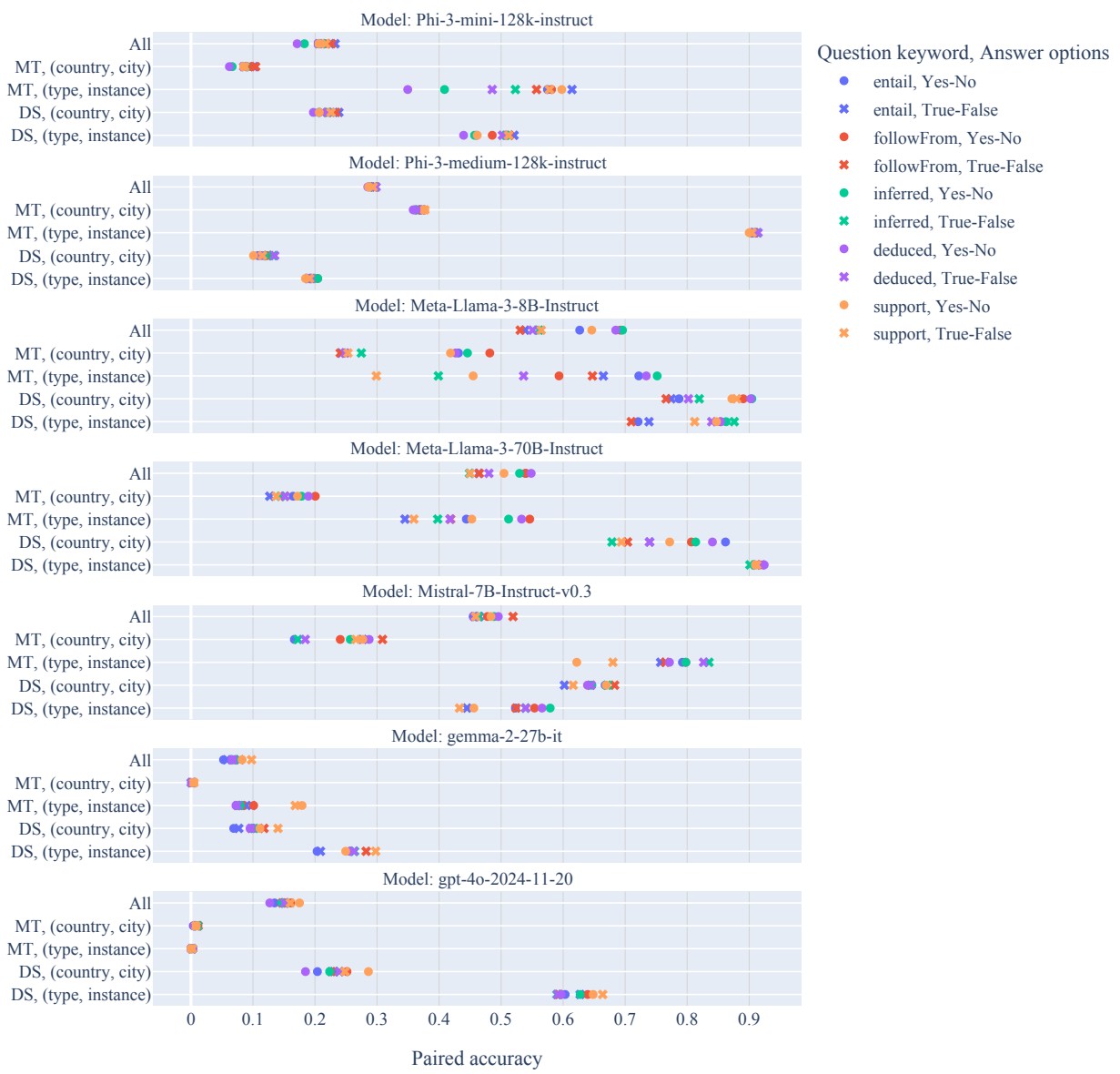

*Figure 5.* Breakdown of results with respect to paired accuracy, by different question phrasings.

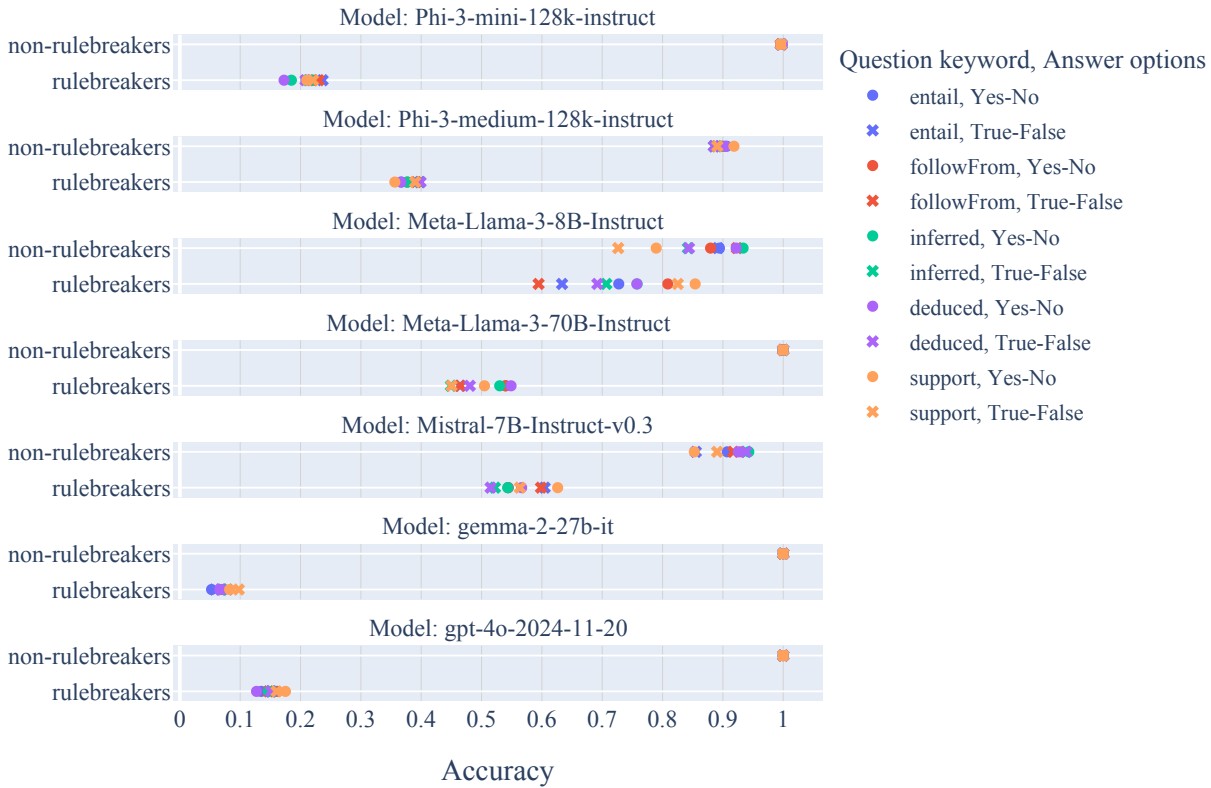

*Figure 6.* Breakdown of results with respect to rulebreaker and non-rulebreaker accuracy, by different question phrasings.

# G. Example Prompts

We list example prompts used in experiment that are produced by applying the chat template specific to each open-weight LLM accessed on Hugging Face. Chat templates are applied using the *apply_chat_template* function as documented in https://huggingface.co/docs/transformers/main/en/chat_templating.

Phi-3-mini-128k-instruct and Phi-3-medium-128k-instruct:

```
<|user|> Premises: If Lív caught some kind of bird, then she did not catch
a goose. Lív caught a goose.\n
Conclusion: Lív did not catch any kind of bird.\n
Do the Premises entail the Conclusion? Answer Yes or No only.<|end|>
<|assistant|>
```

Meta-Llama-3-8B-Instruct and Meta-Llama-3-70B-Instruct:

```
<|begin_of_text|><|start_header_id|>user<|end_header_id|>\n\n
Premises: If Lív caught some kind of bird, then she did not catch a goose.
Lív caught a goose.\n
Conclusion: Lív did not catch any kind of bird.\n
Do the Premises entail the Conclusion? Answer Yes or No only.<|eot_id|>
<|start_header_id|>assistant<|end_header_id|>\n\n
```

Mistral-7B-Instruct-v0.3:

```
[INST] Premises: If Lív caught some kind of bird, then she did not catch
a goose. Lív caught a goose.\n
Conclusion: Lív did not catch any kind of bird.\n
Do the Premises entail the Conclusion? Answer Yes or No only.[/INST]
```

Gemma-2-27b-it:

```
<bos><start_of_turn>user\n
Premises: If Lív caught some kind of bird, then she did not catch a goose.
Lív caught a goose.\n
Conclusion: Lív did not catch any kind of bird.\n
Do the Premises entail the Conclusion? Answer Yes or No only.<end_of_turn>\n
<start_of_turn>model\n
```

## H. Additional Instructions and Variation in Prompt Phrasing

In light of our results in Section 5.1, we further investigate whether prompting instructions that remind LLMs to utilize common sense and factual knowledge when reasoning improve their mediocre performance on RULEBREAKERS.

We do this in two different ways:

1. **Baseline phrasing + additional instructions**. We include the **bold instructions** into each of the same 10 prompt phrasings described in Section 4.2:

   ... [Does the Conclusion follow from the Premises?] **When reasoning, you should take into account your factual knowledge about the real world and assess whether any relevant logical rules should be applied.** Answer [Yes] or [No] only.

2. **Alternative phrasing**. Instead of including additional instructions, we use this alternative question phrasing:

   ... Based on the provided context and commonsense knowledge, is the Conclusion correct? Answer Yes or No only.

Additionally, while using the alternative phrasing, we also prefix the two premises in each prompt with the **bold wording**, to test whether this improves models' ability to recognize the factual contradiction between the two premises (we refer to this condition as **alternative phrasing + prefixed premises**), i.e.:

Premises: **Suppose we are told that** [premise 1]. **However, as a matter of fact,** [premise 2].

Conclusion: [conclusion].

Based on the provided context and commonsense knowledge, is the Conclusion correct? Answer Yes or No only.

As shown in Table 8, using the alternative question phrasing in combination with adding prefixes to the premises improves the paired accuracy achieved by most models[24], with the most substantial gain being +24.13 (%) over the baseline condition for gemma-2-27b-it. However, paired accuracy of all models remain mediocre, ranging from 31.27 (gemma-2-27b-it) to 69.12 (Meta-Llama-3-8B-Instruct).

Importantly, for the majority of the models, we often observe a trade-off in rulebreaker versus non-rulebreaker accuracy under different conditions. For example, while Mistral-7B-Instruct-v0.3 under the **alternative phrasing** condition improves by +5.58 in non-rulebreaker accuracy compared to the baseline condition, the model suffers a drop of -13.62 in rulebreaker accuracy. Conversely, under the **baseline phrasing + additional instructions** condition, Meta-Llama-8B-Instruct improves by +10.51 in rulebreaker accuracy but drops by -18.21 in non-rulebreaker accuracy. We find this "trading off" pattern with respect to Phi-3-mini-128k-Instruct, Phi-3-medium-128k-Instruct and Meta-Llama-3-70B-Instruct. Whilst highlighting the challenge that RULEBREAKERS poses to current LLMs, these results also indicate the inherent tension between prompting models to reason according to what formal logic demands (i.e. robustly applying rules regardless of the semantic content of the propositions) and prompting them to reason in a more human-like and knowledge-informed manner by taking semantic content, factual knowledge and common sense into account.

## I. Task Variation: Conclusion Generation

Inspired by similar experiment setups in Johnson-Laird & Byrne (2002) and Quelhas & Johnson-Laird (2017), instead of presenting both the premises and conclusions to LLMs and asking them to either accept or reject the conclusions (as we had done in our main experiment), we present only the premises and ask the LLMs to generate the conclusions themselves for both rulebreaker and non-rulebreaker scenarios. For example:

Premises: If Lív caught some kind of [*if non-rulebreaker*: insect / *if rulebreaker*: bird], then she did not catch a goose. Lív caught a goose.

*What conclusion, if any, follows from the Premises? If you think nothing follows from the Premises, answer 'Nothing follows'. Keep your response concise.*

---

[24]The evaluation of GPT-4o on the alternative phrasing + prefixed premises condition was carried out in March 2025.

*Table 8.* Comparison of models' accuracy results (paired accuracy ($\tau$), rulebreaker accuracy ($\tau_{DR}$), non-rulebreaker accuracy ($\tau_{DN}$) in %) across four conditions: (a) baseline question phrasings (as described in Section 4.2); (b) baseline question phrasings with additional instructions appended; (c) alternative question phrasing; and (d) alternative question phrasing with prefixed premises. Highest score in each row is highlighted in **bold**. Accuracy gain/loss of each condition (compared against the baseline) is provided in brackets.

| | | BASELINE PHRASING | BASELINE PHRASING + ADDITIONAL INSTRUCTIONS | ALTERNATIVE PHRASING | ALTERNATIVE PHRASING + PREFIXED PREMISES |
|---|---|---|---|---|---|
| PHI-3-MINI-128K-INSTRUCT | $\tau$ | 20.75 | 22.30 (+1.55) | 24.56 (+3.81) | **28.72** (+7.97) |
| | $\tau_{DR}$ | 21.07 | 22.70 (+1.63) | 25.12 (+4.05) | **29.34** (+8.27) |
| | $\tau_{DN}$ | **99.68** | 99.59 (-0.09) | 99.44 (-0.24) | 99.37 (-0.31) |
| PHI-3-MEDIUM-128K-INSTRUCT | $\tau$ | 29.19 | 32.61 (+3.42) | 29.14 (-0.05) | **44.66** (+15.47) |
| | $\tau_{DR}$ | 38.19 | 42.49 (+4.30) | 35.03 (-3.16) | **44.87** (+6.68) |
| | $\tau_{DN}$ | 89.63 | 89.11 (-0.52) | 92.97 (+3.34) | **99.73** (+10.1) |
| META-LLAMA-3-8B-INSTRUCT | $\tau$ | 60.92 | 52.89 (-8.03) | 65.94 (+5.02) | **69.12** (+8.20) |
| | $\tau_{DR}$ | 73.58 | 84.09 (+10.51) | 78.32 (+4.74) | **88.22** (+14.64) |
| | $\tau_{DN}$ | 86.46 | 68.25 (-18.21) | **87.36** (+0.90) | 80.53 (-5.93) |
| META-LLAMA-3-70B-INSTRUCT | $\tau$ | 49.71 | 51.95 (+2.24) | **59.39** (+9.68) | 56.19 (+6.48) |
| | $\tau_{DR}$ | 49.73 | 52.03 (+2.30) | **59.53** (+9.80) | 56.40 (+6.67) |
| | $\tau_{DN}$ | **99.98** | 99.91 (-0.07) | 99.86 (-0.12) | 99.79 (-0.19) |
| MISTRAL-7B-INSTRUCT-V0.3 | $\tau$ | **47.64** | 47.08 (-0.56) | 39.53 (-8.11) | 40.09 (-7.55) |
| | $\tau_{DR}$ | 56.28 | **57.13** (+0.85) | 42.66 (-13.62) | 55.12 (-1.16) |
| | $\tau_{DN}$ | 90.99 | 89.57 (-1.42) | **96.57** (+5.58) | 84.96 (-6.03) |
| GEMMA-2-27B-IT | $\tau$ | 7.14 | 6.99 (-0.15) | 8.65 (+1.51) | **31.27** (+24.13) |
| | $\tau_{DR}$ | 7.14 | 6.99 (-0.15) | 8.65 (+1.51) | **31.28** (+24.14) |
| | $\tau_{DN}$ | **100.00** | **100.00** (+/-0) | **100.00** (+/-0) | 99.99 (-0.01) |
| GPT-4O | $\tau$ | 15.06 | 13.91 (-1.15) | 19.78 (+4.72) | **31.92** (+16.86) |
| | $\tau_{DR}$ | 15.06 | 13.91 (-1.15) | 19.78 (+4.72) | **31.92** (+16.86) |
| | $\tau_{DN}$ | **100.00** | 99.99 (-0.01) | **100.00** (+/-0) | **100.00** (+/-0) |

In the non-rulebreaker case, the answer which we consider correct and expected of typical human reasoners would be "*Lív did not catch an*[25] *insect*", since it is an uncontentious application of *modus tollens*; by contrast, we consider the answer "*nothing follows*" as incorrect.

In the rulebreaker case, the conclusion we consider correct would be "*nothing follows*", as the premises are factually inconsistent with each other (since goose is in fact a type of bird); by contrast, we consider the answer "*Lív did not catch a bird*" as incorrect, since it factually contradicts the second premise. As in our main experiment, this again reflects the views and findings in existing cognitive science literature (Quelhas et al., 2010; Khemlani & Johnson-Laird, 2019, etc.) that, when faced with what they recognize to be inconsistent premises, typical human reasoners are not expected to just conclude that any conclusion whatsoever follows. By contrast, in classical logic, any conclusion follows validly from a set of contradictory premises.

Similarly, for the *disjunctive syllogism* instances, e.g. "*Lív caught either a goose or some kind of [insect/bird]. Lív did not catch [an insect/a bird].*", the correct conclusion would be "*Lív caught a goose*" in the non-rulebreaker case, and "*nothing follows*" in the rulebreaker case.

We prompt each model to generate using greedy decoding and setting a maximum generated token limit of 50 tokens. We then pattern-match their outputs against our expected correct and incorrect answers. Where our pattern-matching fails to map an output to either a correct or incorrect answer, we categorize it as "unparsed".

We compute the frequency of each LLM's correct, incorrect and unparsed responses to all rulebreakers and non-rulebreakers in RULEBREAKERS. Additionally, we apply the same approach in calculating paired accuracy to compute the frequency of correct, incorrect and unparsed responses to rulebreaker and non-rulebreaker *pairs* (see Section 3.4). If a model's response to both the rulebreaker and non-rulebreaker prompt in a pair are correct, we categorize the paired response as correct. If at least one response is unparsed, we categorize the paired response as unparsed. In all other cases, we categorize the paired response as incorrect.

As shown in Figure 7, the results of this additional experiment generally conform to the patterns we had identified in the results of our main experiment in Section 5.1. Most models still perform better with respect to non-rulebreakers than rulebreakers. As in our main experiment, Phi-3-mini-128k-instruct, gemma-2-27b-it and gpt-4o-2024-11-20 exhibit behavior that suggests the models' over-rigidly applying logical rules in generating conclusions, thereby performing particularly well on non-rulebreakers but particularly poor on rulebreakers.

Notable exceptions, however, are Phi-3-medium-128k-instruct and Mistral-7B-Instruct-v0.3. Unlike their behavior in our main experiment, both these models perform better on rulebreakers than non-rulebreakers in this setup. In particular, we observe that Mistral-7B-Instruct-v0.3 exhibits a strong bias towards outputting "*Nothing follows*" which results in a near perfect accuracy on rulebreakers but to the detriment of its performance on non-rulebreakers. As a result, its proportion of correct *paired* responses is even lower than the paired accuracy it achieved in our main experiment. Whilst highlighting an interesting phenomenon that merits future investigation, these results also demonstrate the robustness of our paired accuracy metric against any abrupt shifts in models' preferences due to variations in task and prompt phrasing.

We also conduct a follow-up experiment on a subset of models to validate that our findings above are robust to the use of sampling as opposed to greedy decoding during inference with different temperature settings. As shown in Table 9, the results obtained from using sampling with varying temperatures do not substantially deviate from our baseline. Increasing temperature exhibits mixed effects with respect to different models, harming the performance of Llama-3-8B-Instruct and Mistral-7B-Instruct-v0.3 but improving the performance of Phi-3-mini-128-instruct (which has performed the worst under the baseline condition compared to the other two models).

## J. Further Implementation Details

All open-sourced models used in our study were accessed and downloaded through Hugging Face: `https://huggingface.co/models`.

We ran each model on a single NVIDIA A100 GPU, except for Meta-Llama-3-70B-Instruct which we ran on three.

Running inference on the full RULEBREAKERS dataset permuted by 10 prompt phrasing variations took approximately

---

[25]In extracting the LLMs' answers by pattern-matching, we also allow for the article in this position to be substituted with "any", "any kind of" or "some kind of", since these produce the same meaning.

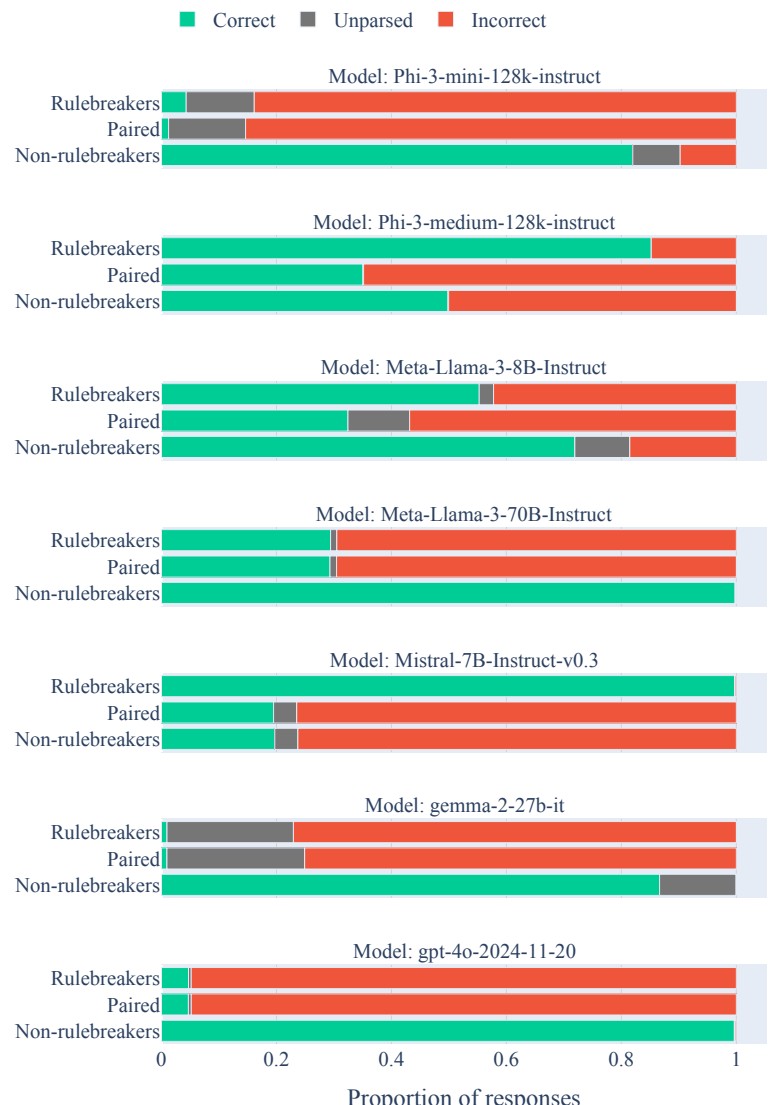

*Figure 7.* Proportion of correct, unparsed and incorrect responses extracted from LLMs' outputs when they are provided with only the premises and instructed to generate conclusions.

*Table 9.* Percentage of corrected paired responses in conclusion generation by using random sampling with different temperatures

| TEMPERATURE | PHI-3-MINI-128K-INSTRUCT | LLAMA-3-8B-INSTRUCT | MISTRAL-7B-INSTRUCT-V0.3 |
|---|---|---|---|
| N/A - GREEDY DECODING (BASELINE) | 1.23 | **32.44** | **19.50** |
| 0.1 | 1.47 | 32.34 | 19.26 |
| 0.5 | 4.24 | 31.76 | 18.68 |
| 1.0 | **7.59** | 28.51 | 17.12 |

one hour per model with batch size of 128, except for Gemma-2-27b-it which took four hours with a batch size of 128, and Meta-Llama-3-70B-Instruct which took 10 hours with batch size of 64.

## K. Further Experiments and Discussion regarding Logic-enhanced Models

While revealing a reasoning blind spot in current general-purpose models, our findings also point to the potential trade-off in methods that rely on formal logic to enhance these models' general reasoning abilities.

For example, intuitively, we would expect that models specifically fine-tuned on logical reasoning datasets would exhibit an even more pronounced tendency to over-rigidly apply formal logic. In addition, some hybrid systems are explicitly guided or constrained to reason according to formal logic. This is done, for example, by using LLMs to parse natural language statements into symbolic logical forms, which are then either processed by an external symbolic solver (as in the case of, e.g., LogicLM (Pan et al., 2023)) or fed back into the model in the subsequent stages as part of a multi-turn prompting strategy (e.g. in SymbolicCoT (Xu et al., 2024)). However, the problem RULEBREAKERS poses to these systems is precisely that each rulebreaker and non-rulebreaker in a pair share the same surface logical form, such that the conclusion is logically valid given the premises in propositional logic. Thus, even if the parsing is correct, the system would theoretically accept the conclusion in both non-rulebreakers (which would be correct) and non-rulebreakers (which would be incorrect), hence leading to low paired accuracy.

To validate these theoretical arguments, we evaluate an existing Llama-3.1-8B-Instruct model on Hugging Face[26] that has been fine-tuned on a dataset for propositional logic reasoning, and compare its performance on RULEBREAKERS against the baseline model before fine-tuning. We also evaluate the performance gpt-3.5-turbo on RULEBREAKERS across four conditions: (a) directly prompting the model for an answer (baseline); (b) using chain-of-thought (CoT) prompting (Wei et al., 2022b); (c) using LogicLM; and (d) using SymbolicCoT.

We prompt the models to answer the question under the **alternative phrasing + prefixed premises** condition introduced in Appendix H, using greedy decoding. With respect to gpt-3.5-turbo, we use the LogicLM authors' implementation (including their default settings)[27] for conditions (a), (b) and (c); and that of the SymbolicCoT authors[28] for condition (d). As in the LogicLM authors' implementation, where the model fails to output valid logical forms that can be successfully executed, it falls back to the answer it outputs under the CoT condition.

As shown in Table 10, the fine-tuned Llama-3.1-8B-Instruct model outperforms the baseline model in accepting and generating the correct conclusions in non-rulebreaker cases. However, as we expected, its performance on rulebreakers is substantially lower than the baseline model, indicating that the fine-tuned model is over-rigidly applying logical rules without regard to the semantic content of the premises.

With respect to gpt-3.5-turbo, Table 10 shows that using LogicLM and SymbolicCoT boosts the model's performance on non-rulebreakers at the expense of its performance on rulebreakers, leading as expected to worse paired accuracy than the baseline. We also observe that using CoT has a similar trade-off in boosting non-rulebreaker performance (to a level comparable to using LogicLM and SymbolicCoT), but the drop in rulebreaker performance is much less drastic. Nonetheless, these results show that the inherent challenge remains in prompting models to reason robustly in a knowledge-informed manner.

## L. Qualitative Analysis

With respect to our hypothesis (1), we observe that the most frequent country-city pairs about which LLMs answer incorrectly tend to be those that are considered less well-known or likely to appear less frequently in the models' training corpora. These include, for example, capital cities of relatively small island countries (such as Funafuti in Tuvalu and Ngerulmud in the Republic of Palau), and those of smaller African countries relative to their neighbours (such as Gitega in Burundi, Bissau in Guinea-Bissau and Maseru in Lesotho). Conversely, the most frequent country-city pairs that these models answer correctly about tend to be those that are relatively better known or larger in size. Examples include New Delhi in India, Madrid in Spain and Tokyo in Japan.

---

[26] https://huggingface.co/ergotts/llama_3.1_8b_prop_logic_ft
[27] https://github.com/teacherpeterpan/Logic-LLM
[28] https://github.com/Aiden0526/SymbCoT

*Table 10.* Results of logic-enhanced models. "Executable subset" refers to the subset of RULEBREAKERS instances where gpt-3.5-turbo has parsed the premises and conclusion into valid logical forms that were successfully evaluated by the symbolic solver. Paired accuracy ($\tau$) for the executable subset is not computed since the model can output valid logical forms for the non-rulebreaker but not the rulebreaker, or vice versa, within a prompt pair.

| | $\tau$ | $\tau_{D^R}$ | $\tau_{D^N}$ |
|---|---|---|---|
| LLAMA-3.1-8B-INSTRUCT | | | |
|   BASELINE MODEL | **60.68** | **65.68** | 94.67 |
|   FINE-TUNED WITH LOGIC DATASET | 19.03 | 20.67 | **97.15** |
| | | | |
| GPT-3.5-TURBO | | | |
|   BASELINE MODEL | 50.80 | **90.88** | 59.92 |
|   COT (2-SHOT) | **65.36** | 71.78 | 91.75 |
|   LOGICLM | 30.81 | 32.56 | **94.59** |
|     (EXECUTABLE SUBSET) | - | 5.98 | 95.07 |
|   SYMBOLICCOT | 9.53 | 10.47 | 91.33 |

Interestingly, LLMs seem to do particularly well on cities that also contain their country names (e.g. Mexico City in Mexico, Kuwait City in Kuwait). We speculate that this might be because these entity pairs make it easier for models to recognize the implicit contradiction between the factual premise and the conclusion in rulebreakers on a symbolic level (i.e. between "*Anne is in [XYZ] City*" and "*Anne is not in [XYZ]*").

Furthermore, this "symbolic overlap" factor may also hint at a possible explanation for the trend we observed in Section 5.1 that models generally perform better on the DS subset rather than the MT subset of RULEBREAKERS. Intuitively, for a model to apply a logical rule, it needs to recognize where segments of the natural language premises are negations of one another and abstract these segments accordingly e.g. into "*B*" and "*not B*". This mapping is arguably more obvious and straightforward in MT cases than in DS ones. For example, in MT (country, city) cases, the **bold** and underlined parts form a clear negation pair:

"If Anne is in Sweden, then **she is not in Stockholm**. Anne is in Stockholm." (*If A, then not B. B.*)

By comparison, the negation pair is less obvious in DS (country, city) cases:

"**Anne is** either in Stockholm or **somewhere in Sweden**. Anne is not in Sweden." (*A or B. Not B.*)

We conjecture that this may make it more difficult for models to map DS premises and conclusions to their logical forms, compared to MT ones. As a result, models may be less likely to rigidly apply logical rules, hence avoiding incorrect conclusions in DS rulebreaker cases. Nevertheless, as with other factors such as entity familiarity, the precise impact of this "structural-similarity effect" on model behavior may vary across models, potentially contributing to the performance variation we observe.

## M. Unnormalized Score Ratios

Table 11 shows a variation of Table 4 that lists the importance score ratios computed without normalizing by the number of tokens aggregated in each of the two premises. As expected, since the second premise always contains fewer tokens than the first, the score ratios listed in Table 11 are all lower than their counterparts in Table 4. Nonetheless, we do not observe any substantive deviations from the patterns we have identified with respect to the normalized score ratios in Table 4. This indicates that our findings are not substantially affected by variations in how prompts in our experiment are tokenized for each model.

## N. Counting and Comparing Activated Neurons in Rulebreakers versus Non-rulebreakers

To better understand how LLMs internally distinguish rulebreakers from non-rulebreakers, we perform an additional experiment inspired by Voita et al. (2024), focusing on individual neurons' activation values between the two linear layers of the feedforward neural network (FFN) within each model layer's Transformer block.

*Table 11.* Mean unnormalized score ratios (with sd) of each LLM. The paired accuracy ($\tau$) from Section 5.1 is provided for the analysis. The greater the ratio, the more the model attends to the factual context.

| | SCORE RATIO (INPUT $\times$ GRADIENT) | SCORE RATIO (ATTENTION) | $\tau$ |
|---|---|---|---|
| PHI-3-MINI-128K-INSTRUCT | 0.412 (0.152) | 0.572 (0.125) | 0.208 |
| PHI-3-MEDIUM-128K-INSTRUCT | 0.358 (0.090) | 0.589 (0.320) | 0.292 |
| META-LLAMA-3-8B-INSTRUCT | 0.441 (0.109) | **0.991** (0.184) | **0.609** |
| META-LLAMA-3-70B-INSTRUCT | **0.445** (0.119) | 0.877 (0.165) | 0.497 |
| MISTRAL-7B-INSTRUCT-V0.3 | 0.405 (0.111) | 0.771 (0.169) | 0.476 |
| GEMMA-2-27B-IT | 0.412 (0.095) | 0.815 (0.116) | 0.071 |

Specifically, given an input prompt, we count the number of activation values that are greater than zero[29], summed over all token positions, with respect to each layer within a model.

We divide all prompts used in our main experiment described in Section 4.2 into four categories: rulebreakers versus non-rulebreakers that an LLM answered correctly versus incorrectly. We then compute the average number of activated neurons across all prompts within each of the four categories.

As the results in Figure 8 show, models differ widely in their activation counts with respect to each of the four categories. On one end of the spectrum, we see a near perfect overlap in activated counts across all layers in Gemma-2-27b-it – in other words, there is no substantial difference in the average number of neurons activated by rulebreaker or non-rulebreaker prompts (regardless of whether or not the model has answered the prompt correctly). On the other end, with respect to Meta-Llama-3-70B-Instruct, we observe that the four categories of prompts are clearly (and consistently across all layers) distinguishable in terms of activated counts. Somewhat surprisingly, in three out of six models (Phi-3-mini-128k-instruct, Meta-Llama-3-8B-Instruct and Meta-Llama-3-70B-Instruct), incorrectly answered rulebreakers and non-rulebreakers appear to activate more neurons than correctly answered ones. These results suggest a non-trivial relationship between a model's neuron activations and its responses to rulebreakers and non-rulebreakers, presenting an avenue for future investigation.

---

[29]We select zero as a threshold for consistency (whilst recognizing that these values can also be less than zero), since Gemma-2-27b-it uses GeLU activation function estimated with tanh, whereas other LLMs in our study use SiLU (Hendrycks & Gimpel, 2023).

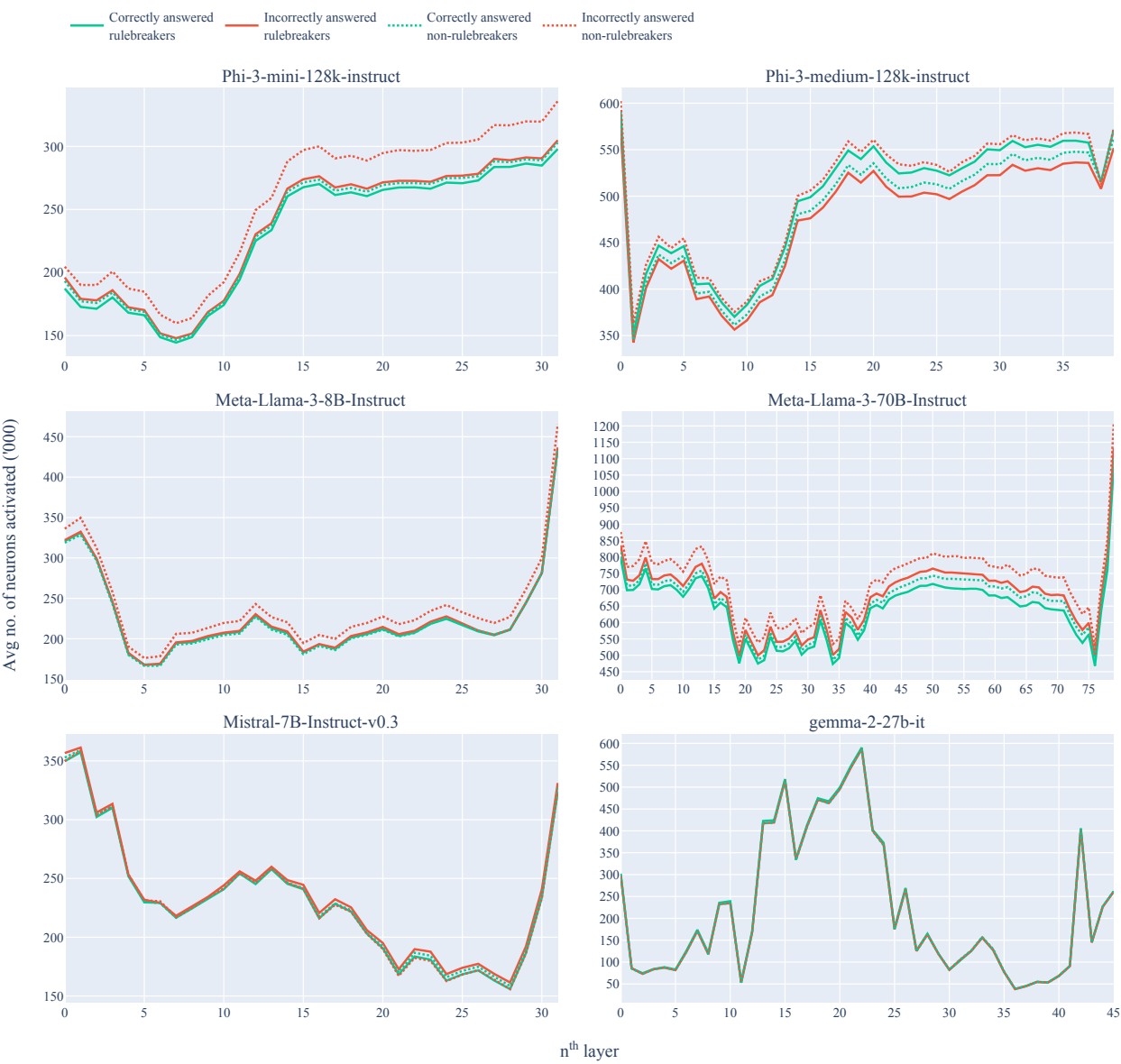

*Figure 8.* Average number of neurons activated per model layer, with respect to prompts belonging to each of four categories (rulebreaker vs non-rulebreaker, correctly vs incorrectly answered by each model).

