# OpenReview forum: "RULEBREAKERS: Challenging LLMs at the Crossroads between Formal Logic and Human-like Reasoning"
_ICML.cc/2025/Conference — ICML 2025 poster_

### Official Review · Reviewer_8kkH · 2025-02-28

**Overall Recommendation:** 1

**Summary:**

The paper RULEBREAKERS: Challenging Large Language Models at the Crossroads between Formal Logic and Human-like Reasoning introduces RULEBREAKERS, a dataset designed to evaluate large language models' (LLMs) ability to distinguish between logical rule-based conclusions and conclusions that align with human reasoning, which incorporates commonsense and factual knowledge. The study defines "rulebreakers" as scenarios where conclusions derived using formal logic contradict human expectations. Evaluating seven state-of-the-art LLMs, including GPT-4o, the paper finds that most models perform poorly on recognizing rulebreakers, often over-applying formal logic in a rigid manner. The authors identify two possible reasons for this failure: (1) models' poor utilization of world knowledge, and (2) suboptimal attention allocation in reasoning. Their findings highlight a crucial limitation of LLMs and provide a counterpoint to recent works that integrate formal logic to improve LLM reasoning, warning against potential divergences from human-like reasoning.

**Claims And Evidence:**

The claims in RULEBREAKERS: Challenging Large Language Models at the Crossroads between Formal Logic and Human-like Reasoning are largely supported by empirical evidence.

**Essential References Not Discussed:**

https://arxiv.org/pdf/2307.02477: Reasoning or Reciting? Exploring the Capabilities and Limitations of Language Models Through Counterfactual Tasks

https://arxiv.org/pdf/2207.07051: Language models show human-like content effects on reasoning tasks

**Experimental Designs Or Analyses:**

The experimental design and analyses in RULEBREAKERS appear to be thoughtfully constructed, with several mechanisms to ensure validity.

**Methods And Evaluation Criteria:**

Yes, the methodology is well-designed for assessing how LLMs handle a crucial failure mode: rigidly applying logical rules without considering common sense or world knowledge. The dataset ensures control over semantic content while allowing systematic testing across models. The evaluation metrics provide both direct (accuracy) and indirect (confidence, attention) insights into model reasoning.

**Other Comments Or Suggestions:**

The naming of RULEBREAKERS vs. non-rulebreakers is clear, but a brief mention of alternative terminology (e.g., "rigid logic contradictions") could help connect the paper to related work in logic and commonsense reasoning.

**Other Strengths And Weaknesses:**

The main finding of this paper seems to have already been well-studied by previous works:

https://arxiv.org/pdf/2307.02477: Reasoning or Reciting? Exploring the Capabilities and Limitations of Language Models Through Counterfactual Tasks

https://arxiv.org/pdf/2207.07051: Language models show human-like content effects on reasoning tasks

**Questions For Authors:**

Are the authors aware of these two well-established works that study very similar topics and has come up with very similar conclusions?

https://arxiv.org/pdf/2307.02477: Reasoning or Reciting? Exploring the Capabilities and Limitations of Language Models Through Counterfactual Tasks

https://arxiv.org/pdf/2207.07051: Language models show human-like content effects on reasoning tasks

**Relation To Broader Scientific Literature:**

The RULEBREAKERS paper is well-situated within the broader scientific literature at the intersection of formal logic, human-like reasoning, and LLM evaluation. It contributes to multiple strands of prior research, including cognitive science, natural language inference, logic-based AI methods, and LLM reasoning evaluation.

**Theoretical Claims:**

There is no theoretical claim.

---

> ### Author Rebuttal · Authors · 2025-04-01
>
> We thank Reviewer 8kkH for commending our “**methodology is well-designed**”, our claims are “**largely supported by empirical evidence**”, and that our experimental design and analyses are “**thoughtfully constructed, with several mechanisms to ensure validity**”. We further appreciate and agree with their assessment that our paper is “**well-situated within the broader scientific literature**” and “**contributes to multiple strands of prior research**”.
>
> **Weakness/Question: “finding well-studied by previous works” [1, 2]**
>
> The only weakness raised by the Reviewer may stem from a misunderstanding as the objective, methodology, and findings of these two works are fundamentally different from ours. In light of the clarifications to follow and the Reviewer’s positive comments above, we respectfully ask the Reviewer to consider whether an adjustment to their initial score may be appropriate.
>
> We kindly refer to **Appendix A (lines 829-833), where we differentiate our work from Lampinen et al. (2024) [1]** and the comparison table below. We did not initially include Wu et al. (2024) [2] because their methodology (applying a counterfactual ontology to logical templates) and findings have largely been established in **Saparov and He (2023) [3]**, which we also already distinguished in **Appendix A (lines 846-852)**. We are grateful for the suggestion and will add more discussion in the final version to make it clear.
>
> |  | **Example instance in dataset** | **Dataset uses a counterfactual ontology?** | **Conclusion is valid according to logic?** | **Premises are not true in the real world?** | **Conclusion is not true in the real world?** | **Assuming the premises are true, does the conclusion contradict any of the Premises?** | **Relevant finding** |
> |---|---|---|---|---|---|---|---|
> | Lampinen et al. (2024) [1] | Premises: All Swedish cities are French cities. All French cities are in Poland. Conclusion: All Swedish cities are in Poland. | Yes to some extent. Examples that violate real-world knowledge were manually written. | Yes | Yes. Sweden, France and Poland are three separate countries, with different cities. | Yes | No | When performing **logical reasoning**, LLMs tend to be biased in judging conclusions that are true in the real world as logically valid, and conclusions that are false in the real world as invalid, even where the underlying logical structure is the same in both cases. |
> | Wu et al. (2024) [2] | Premises: Swedish cities are French cities. French cities are American cities. Conclusion: Swedish cities are American cities. | Yes | Yes | Yes. As above, Sweden, France and America are three separate countries, with different cities. | Yes | No | In a **logical reasoning** task, the more the premises deviate from real-world knowledge, the worse LLMs tend to perform in correctly applying logical rules to derive a valid conclusion. |
> | Our work | Premises: Anne is either in Stockholm or somewhere in Sweden. Anne is not in Sweden. Conclusion: Anne is in Stockholm. | No | Yes | Undetermined | Undetermined | Yes. In the example rulebreaker, the conclusion contradicts the second premise because it is factually impossible for Anne to be in Stockholm if she is not in Sweden, given that Stockholm is located in Sweden in the real world. | When **reasoning with natural language _in general_**, LLMs tend to accept conclusions that can be derived by rigidly applying logical rules, even when these conclusions contradict the premises in fact. |
>
> As shown in the table above, our work differs significantly from [1] and [2] in that (a) we do not use a counterfactual ontology, (b) we do not introduce premises or conclusions that are untrue in the real world; and (c) our findings are not concerned with models’ performance in purely logical reasoning tasks, but with their ability to reason in general with natural language.
>
> As the same two papers are mentioned in the Reviewer's question, we avoid repeating our response here for clarity.
>
> **Suggestion: alternative terminology**
>
> We thank the Reviewer for suggesting alternative terminology to connect our paper to other related work. We will incorporate this by describing rulebreaker cases as “factual contradictions arising from over-rigid reasoning with formal logic” in the final version. Additionally, we will make clear the connection and implications of our findings for LLM applications in knowledge-intensive tasks as we discuss in our S1 response to Reviewer twqU.
>
> [1] Wu et al. https://aclanthology.org/2024.naacl-long.102
>
> [2] Lampinen et al. https://doi.org/10.1093/pnasnexus/pgae233
>
> [3] Saparov and He. https://openreview.net/pdf?id=qFVVBzXxR2V

---

### Official Review · Reviewer_fmAM · 2025-03-04

**Overall Recommendation:** 3

**Summary:**

The authors propose a new dataset for single step reasoning, which consists of pairs of premises and a conclusion, which are answered in a binary way. The premise and the conclusion always are true if one only follows the logical reasoning. However the pair is divided into "rulebreakers", where the reasoning contradicts common knowledge, and "non-rulebreakers", where the reasoning is consistent with such knowledge. The dataset consists of 12,800 of such pairs, which generated based on a template, that the authors define in their paper.

The authors then proceed to evaluate six open source large language model as well as GPT-4o. They evaluate their accuracy on getting the pairs completely correct as well as their performance on the individual sets. They also conduct experiments with two different types of logical reasoning and the model confidence in their outputs for the open source models.

The authors conduct an analysis in two areas. Firstly, whether the performance of the models depends to a certain degree on their familiarity with the respective area of knowledge required for a given question, where they find the key insight that a model might be excellent at knowledge retrieval and/or logically reasoning, but can not recognize that a conclusion is inconsistent with factual knowledge. They also investigate whether models pay attention to the right information in the premises and here the insights are not conclusive, but in general the models that pay more attention to the factual information in the second premise also produce better results overall.

## update after rebuttal

After a cursory reading of the two papers that reviewer 8kkH suggested, I agree with him, that they reduce the novelty of the presented approach and therefore also reduce my score to 3. However the authors have written very clearly how they construct their dataset, which is the main focus of their work. They also use a templated dataset, which makes it harder for language models to just learn the answers and also more extensible. Their evaluation shows the benefits of their dataset. The other papers in comparison rely on small datasets and/or manually written datapoints. I therefore think their paper deserves a fair chance and disagree with the negative view of reviewer 8kkH.

In my opinion, the main focus of the article is the dataset, which the authors describe in detail and their description is well-written and easy to follow. So even if the authors can not fully explain the trends, as reviewer QkHy remarks, seen in their analysis, the evaluation shows the necessity for such a dataset.

**Claims And Evidence:**

Mostly yes.

If I understand Figure 6 correctly, it is based on token probability of the first token in the country name. However the results include GPT-4o, a closed source model, where the authors previously claimed that they cannot access such information.

The authors claim that most humans would recognize these "rulebreakers", however there is no evaluation data provided in the paper itself.

**Essential References Not Discussed:**

No, not to my knowledge.

**Experimental Designs Or Analyses:**

Evaluation methodology seems simple enough and reasonable. Prompts and samples are documented well enough. The conducted analyses seems convincing, albeit sometimes inclusive.

**Methods And Evaluation Criteria:**

Yes, the authors created their own dataset and evaluate various aspects of that dataset for seven different models.

**Other Comments Or Suggestions:**

typos:
* line 395: extra white space between end of the sentence and the footnote number
* line 832: "all As and Cs" - typo: "and" -> "are"

**Other Strengths And Weaknesses:**

strengths:
* paper was clearly written, well presented and easy to follow
* well executed and broad evaluation

weaknesses:
* idea is rather simple and only based on a few templates, which begs the question
* only single step reasoning, but even so models show only poor or mediocre performance

**Questions For Authors:**

* How can GPT-4o be in Figure 6, if the calculation relies on the token probability?
* Did you consider providing a baseline based on humans solving these questions?

**Relation To Broader Scientific Literature:**

The authors propose a new dataset for testing, whether language models can conduct reasoning, but also realizes if the reasoning conflicts with factual knowledge. While the idea for the dataset is simple, that idea is presented clearly and the description is easy to follow, which should make the use of the dataset fairly straightforward.
The authors intend to contribute to a growing body of literature, that cautions the increased use of reasoning by language models.

**Theoretical Claims:**

There are no theoretical claims. The authors define the metrics for their evaluation with formulas, which look reasonable.

---

> ### Author Rebuttal · Authors · 2025-04-01
>
> We thank Reviewer fmAM for their helpful feedback. We appreciate their positive comments that our evaluation is “**well executed and broad**”, our analyses are “**convincing**” and that our paper is “**clearly written, well presented and easy to follow**”.
>
> **Q1: “How can GPT-4o be in Fig 6, if calculation relies on token probability?”**
>
> We can only access probabilities of the top 20 most likely tokens predicted by GPT-4o. The Reviewer is correct that Fig 6 is based on probability of the first token in the country name. By design, our experiment’s control for factual knowledge (**lines 257-278**) **guarantees** that the first token in the country name is **the most likely token** predicted by the model at that timestep. We can therefore access its predicted probability. By contrast, our method for model confidence (**lines 251-267**) requires access to probability of tokens which may not be in the top 20 predictions.
>
> **Suggestion 1 (S1): “claim that humans recognize 'rulebreakers', but no evaluation data provided”**/**Q2: “Did you consider providing a baseline based on humans solving these questions?”**
>
> Our dataset is based on rulebreakers used in prior cognitive science studies (see table below), which were all manually **crafted by expert psychologists and tested on human participants under controlled settings**. These studies already establish a baseline, finding that people generally handle these rulebreakers as we expected. Our key contribution is in systematically scaling these examples by identifying recurring patterns, creating templates and generating controlled variations that preserve the structure of these initial examples.
>
> Also, **the ability to recognize rulebreakers is contingent on reasoners having the relevant knowledge (e.g. that “Stockholm is in Sweden”)**. Since LLMs have often been observed to possess broader factual knowledge, directly comparing their performance on RULEBREAKERS against humans would not be valid. As such, we consider that an additional human baseline would add limited value. We are happy to conduct a small-scale human annotation but, given limited time to hire annotators, will include it in the final version.
>
> | **Study** | **Examples from dataset** | **Our corresponding rulebreaker template(s)** | **No. of participants in study** | **Relevant finding** |
> |---|---|---|---|---|
> | Quelhas et al. (2010) [1] | “If Manuel plays a game, then he doesn’t play football” | MT (type, instance) | 28 | Participants generally avoid conclusions that factually contradict the premises, even where they can be derived by applying a logical rule (modus tollens). Instead, they conclude that “nothing follows” from the premises. |
> | Quelhas and Johnson-Laird (2017) [2] | “Andre is in Lisbon or he is in Portugal”; “Luis is eating chicken or he is eating meat” | DS (country, city), DS (type, instance) | 80 | Same as above, with respect to the logical rule of disjunctive syllogism. |
> | Johnson-Laird and Byrne (2002) [3] | “If Bill is in Brazil then he is not in Rio de Janeiro”; “If Ann is in the Hotel LaBlanc then she is not in the Champagne Suite” | MT (country, city) | 41 | When participants are familiar with the entities mentioned in the premises, they are more likely to recognize and avoid factually contradicting conclusions, as compared to when they are not. |
>
> **Weakness 1 (W1): “idea is rather simple and only based on a few templates”**
>
> Our approach is intentionally grounded on existing cognitive science studies: we ensure the **four** templates (from which we generated **25,600 instances**) are directly based on examples carefully crafted and validated in prior work with human participants.
>
> Expanding the set of templates is a promising direction for future work but, as the Reviewer rightly implies, such new templates will also need to be based on examples that are validated empirically with human participants. We hope our novel methodology and findings will inspire efforts in this direction from both NLP and cognitive science communities.
>
> **W2: “only single step reasoning”**
>
> As the Reviewer rightly points out, models exhibit poor to mediocre performance even in single-step settings. We believe that a targeted and rigorous study of single-step inferences is critical in exposing blind spots that lead to failures in multi-step settings.
>
> While we specifically focus on inherent patterns of a model’s reasoning process independent of any prompt engineering, future work could design prompting techniques to steer model behaviour and address the blind spot we identified of LLMs over-rigidly applying logical rules to accept and draw factually inconsistent conclusions.
>
> **S2: typos**
>
> Thank you for spotting these. We will incorporate the corrections in our final version.
>
> [1] Quelhas et al. https://doi.org/10.1080/17470210903536902
>
> [2] Quelhas and Johnson-Laird. https://doi.org/10.1080/17470218.2016.1154079
>
> [3] Johnson-Laird and Byrne. https://doi.org/10.1037/0033-295X.109.4.646

---

### Official Review · Reviewer_QkHy · 2025-03-13

**Overall Recommendation:** 2

**Summary:**

This paper introduces RULEBREAKERS, a dataset specifically created to assess LLMs on reasoning scenarios that emphasize "human-like reasoning" over logic reasoning. The study demonstrates that state-of-the-art LLMs, including GPT-4o, frequently apply logical rules, which is inconsistent with human reasoning.

**Claims And Evidence:**

No. "Human reasoning" is undefined and highly ambiguous. A person with even basic logic training would not find the example in Figure 1 counterintuitive. Moreover, while the dataset is claimed to evaluate human-like reasoning, it was neither created nor validated by humans, making the claim unsupported.

**Essential References Not Discussed:**

No

**Experimental Designs Or Analyses:**

Yes

**Methods And Evaluation Criteria:**

No. See above.

**Other Comments Or Suggestions:**

1. Potential effects of prompt phrasing on model performance are recognized but not deeply explored.
2. The hypothesis regarding model over-generalization of logic rules is plausible but could benefit from further empirical validation.

**Other Strengths And Weaknesses:**

Strengths:
Extensive compared multiple advanced LLMs.

Weaknesses:
1. "Human reasoning" is undefined and highly ambiguous. A person with even basic logic training would not find the example in Figure 1 counterintuitive. Moreover, while the dataset is claimed to evaluate human-like reasoning, it was neither created nor validated by humans, making the claim unsupported.
2. Limited exploration into internal model mechanisms that cause failure.

**Questions For Authors:**

1. How to define "human-reasoning" and why?
2. Can you provide additional details on how model familiarity with entities impacts performance?
3. Have experiments been considered with models specifically fine-tuned for logical reasoning tasks?

**Relation To Broader Scientific Literature:**

This work tries to connect cognitive science and LLM reasoning.

**Theoretical Claims:**

N/A

---

> ### Author Rebuttal · Authors · 2025-04-01
>
> We thank Reviewer QkHy for their helpful feedback and recognizing that our study was “**extensive**” in having “**compared multiple advanced LLMs**”.
>
> **Weakness 1 (W1): “human reasoning is undefined”/Question 1 (Q1): “how to define human-reasoning and why?”**
>
> We will **replace references to “human reasoning” with “knowledge-informed reasoning”**.
>
> We define reasoning to be “knowledge-informed” if a reasoner incorporates factual and commonsense knowledge in their reasoning process, and does not draw or accept conclusions that factually contradict any premises. Accordingly, a model that accepts the conclusions in our rulebreaker cases would fail to be “knowledge-informed”.
>
> Our definition is motivated by arguments in existing cognitive science literature that humans typically incorporate factual and commonsense knowledge to interpret natural language premises, and avoid drawing conclusions that, to their knowledge, factually contradict any premises. We discussed these in **lines 45-63 and 96-116** and will add more justifications for our definition.
>
> This term also highlights our findings' implications for LLM use in knowledge-intensive tasks, as discussed in our S1 response to Reviewer twqU.
>
> **W2: “example in Fig 1” is not counterintuitive**
>
> The example is commonly used in cognitive science works (e.g. [1], [2]) to demonstrate reasoning problems with “relevance conditional” statements. We will replace this with an example from our own dataset to better represent our rulebreakers.
>
> **W3: “dataset...was neither created nor validated by humans”**
>
> We kindly refer to our S1/Q2 response to Reviewer fmAM.
>
> **W4: “Limited exploration into internal model mechanisms that cause failure”**
>
> Please see **Appendix M**: we analyzed neuron activations in feedforward layers, complementing two potential causes diagnosed in Section 6. We welcome further suggestions on specific investigation methods.
>
> **Suggestion 1 (S1): “Potential effects of prompt phrasing…not deeply explored.”**
>
> We refer to **Appendices D and F**: we analyzed the breakdown of our results by different phrasings, and tested more re-phrasing and additions to confirm that our findings are robust against prompt variations.
>
> **S2: “hypothesis regarding model over-generalization …could benefit from further empirical validation”**
>
> See **Appendices F and G**: we validated our main results with further experiments, in addition to existing controls for models’ factual knowledge and prompt sensitivity (as above).
>
> **Q2: “Can you provide additional details on how model familiarity with entities impacts performance?”**
>
> To clarify **lines 396-416**, we found that a model generally performs worse on prompts containing unfamiliar entities, as compared to prompts containing familiar ones. This is supported by our qualitative analysis in **Appendix I**. **This trend mirrors findings from cognitive science [3]** that when participants are familiar with entities in the premises, they are more likely to recognize and avoid factually contradicting conclusions, as compared to when they are not.
>
> Separately, when comparing across models, we found that simply being familiar with entities does not guarantee good performance on RULEBREAKERS: while Gemma-2-27b-it is overall highly familiar with entities in our dataset, it scores among the poorest in paired accuracy. This shows that **a model can excel at recalling factual knowledge yet still struggle to apply that knowledge effectively in reasoning**. We will add this expanded explanation in the final version.
>
> **Q3: “Have experiments been considered with models specifically fine-tuned for logical reasoning tasks?”**
>
> We did not do so, as our work initially aimed to highlight a blind spot that exists even in general-purpose models. We expect the behaviour of over-rigidly applying logic to be even more pronounced in such fine-tuned models.
>
> To test this, we **evaluate a Llama-3.1-8B-Instruct** on Hugging Face [4] **fine-tuned on a dataset for propositional logic reasoning**, and compare its performance on RULEBREAKERS against the baseline model before fine-tuning.
>
> |  | **Main experiment setup (% accuracy)** |  |  | **Appendix G setup** |  |
> |---|---|---|---|---|---|
> |  | Paired  | Rulebreakers | Non-rulebreakers | % of correct conclusions generated in rulebreaker cases | % of correct conclusions generated in non-rulebreaker cases |
> | Baseline | **50.45** | **91.46** | 58.60 | **74.20** | 47.91 |
> | Fine-tuned | 11.42 | 32.93 | **75.48** | 0.40 | **65.75** |
>
> As expected, the fine-tuned model performs better on non-rulebreakers but substantially worse on rulebreakers compared to the baseline. We will include a detailed discussion of these results in the final version.
>
> [1] Johnson-Laird. Mental models. Cambridge University Press, 1983.
>
> [2] Quelhas et al. https://doi.org/10.1080/17470210903536902
>
> [3] Johnson-Laird and Byrne. https://doi.org/10.1037/0033-295X.109.4.646
>
> [4] huggingface.co/ergotts/llama_3.1_8b_prop_logic_ft

---

### Official Review · Reviewer_twqU · 2025-03-14

**Overall Recommendation:** 3

**Summary:**

The authors introduce RULEBREAKERS, a dataset designed to assess LLMs' ability to reason using common sense and factual knowledge rather than strictly following formal logic. The experimental evaluation proposed in the paper spans over seven LLMs, and its findings uncover a notable weakness in these models' ability to identify and reject conclusions derived from propositional logic that are factually inconsistent with the premises. The paper is primarily well-written, easy to understand, and follow (excluding a few paragraphs). The results are interesting and highlight an open challenge that LLMs need to face. Meanwhile, the experimental settings could be extended to take into account large LLMs and hybrid approaches to test the impact of model complexity/size on its reasoning capability. Overall, the paper feels relevant. Thus, I consider the paper slightly above the ICML conference's acceptance threshold.


## update after rebuttal

I thank the authors for the clear and detailed rebuttal they provided. Overall, I think the authors covered some of my doubts, while leaving also some open questions. Therefore, I am considering the paper to be generally solid and my feeling concerning it is still positive. Therefore, I am keeping my original review score (which was already positive).

**Claims And Evidence:**

The authors’ claims are largely supported by empirical evidence. Although the experimental evaluation could be extended to take into account larger LLMs or hybrid approaches, the performance of the tested models seems to back the authors’ claims in most experiments. Therefore, I believe that the authors' claims are sound.

**Essential References Not Discussed:**

I don’t believe there are any essential references missing. However, given that I’m not an expert of the topic, I might be wrong.

**Experimental Designs Or Analyses:**

The evaluation of the experiments proposed by the authors is sound and extensive enough to back up their claims. The rule-breakers are tested over a few different LLMs, thus giving an overview of general LLM limitations. However, the experimental evaluation could still be extended to larger LLMs to identify if the limitations highlighted by the authors are connected to model size or if an intrinsic issue exists with how the LLM thought process is brought about.

**Methods And Evaluation Criteria:**

The methodology proposed by the authors to test the performance of LLMs on rule-breakers is reasonable. The evaluation metrics considered highlight different aspects of the LLM's capabilities, allowing the authors to identify a few different relevant challenges related to rule-breakers and LLMs. Overall, the methods and evaluation criteria are reliable.

**Other Comments Or Suggestions:**

- Although I understand that the model selection process was bounded by resource limitations, leveraging only small/medium-sized LLMs for the experimental evaluation represents a relevant issue for the paper. Indeed, it would be interesting to understand if larger, more complex models are capable of performing better on rule-breakers. As it stands, the experimental evaluation is a bit undersized with respect to the relevance of the task considered.
- The temperature parameter used by the LLM represents a very relevant hyper-parameter to study for validating the output. However, the authors fail to mention anything about such a parameter in the paper (unless I failed to read it properly). The behaviour of an LLM when varying the temperature parameter may largely vary, thus altering the performance of the obtained output. Therefore, I would suggest the authors add some other experiments to show what happens at the variation of the temperature.
- Looking at Figure 3, it seems that the performance of the same LLM largely varies across the different types of rule-breakers. Also, there does not seem to be a precise correlation between the performance of the different models on the different types of rule breakers. For example, Phi3 medium performs much better on MT (type, instance) than on DS (country, city). Meanwhile, Llama3 70B is the exact opposite. However, the authors seem to fail to mention anything concerning this aspect in their discussion of the results. Therefore, I suggest the authors add more insights concerning why these behaviours emerge.
- Figure 5 is not presented clearly. In my opinion, section 5.2 is a bit tricky to follow as the authors confused the readers when presenting the results. I suggest the authors rephrase it a bit and be more careful in their writing.
- The authors mention, “If our hypothesis (1) is correct, we would expect models to have a higher “familiarity” with respect to prompt pairs in their “recognized” group, i.e. those that the model has answered correctly. As shown in Figure 6, this holds true for all LLMs except for Meta-Llama-3-8B-Instruct.”. However, from Figure 6, it seems like the opposite is true. This may be due to the authors plotting the results in the form of a box plot, which does not allow us to distinguish the nuanced variations among the red and green boxes well. From an outside perspective, it seems like the boxes are very much overlapping, thus invalidating the authors' statement. Assuming that the authors’ statement is indeed true, I would suggest that the authors find a better representation of the results in Figure 6. A simple table with a mean and standard deviation of the familiarity would be enough.
- Since the authors mentioned that there exist approaches incorporating logic-based training metrics or constraints during inference in LLMs, it would be interesting to check how these hybrid models perform in the rule-breakers dataset.

**Other Strengths And Weaknesses:**

STRENGTHS:
- The paper tackles an interesting research field, analyzing the ability of LLM to reason in a human-like manner
- The rue-breakers proposed by the paper seem to be constructed in a sound manner
- The obtained dataset is extensive and represents a valid addition to the LLM community
- The experimental evaluation highlights some interesting findings on LLM’s behaviour on such rule-breakers

WEAKNESSES:
- Some findings are not clearly presented by the authors. Such as section 5.2 or the claim on the familiarity effect in section 6.
- The authors selected only small/medium-sized LLMs. Thus, the validity of the paper’s findings may be limited.
- The authors mention that approaches exist that incorporate logic-based training metrics or constraints during inference in LLMs. Still, they do not check how these hybrid models perform in the rule-breaker settings.

**Questions For Authors:**

- Could the authors discuss what they would expect to find out when applying the same experimental evaluation to larger models?
- Did the authors consider the possible impact of the temperature parameter when prompting the LLMs in their experimental evaluation?
- Looking at Figure 3, the performance of the same LLM largely varies across the different types of rule-breakers. Do the authors have any insight on why this happens? Also, there does not seem to be a precise correlation between the performance of the different models on the different types of rule breakers. For example, Phi3 medium performs much better on MT (type, instance) than on DS (country, city). Meanwhile, Llama3 70B is the exact opposite. Do the authors have any insight on why this is happening?
- Could the authors provide additional experiments implementing hybrid LLM approaches incorporating logic-based training metrics or constraints during inference to test their performance on the rule-breakers dataset?

**Relation To Broader Scientific Literature:**

The authors did not mention how the failure of LLMs to recognize and handle rule-breakers could impact their application from a real-world perspective.

**Theoretical Claims:**

The authors did not provide any theoretical claims given that the paper focuses on experimental analysis only. Therefore, this question is not applicable.

---

> ### Author Rebuttal · Authors · 2025-04-01
>
> We thank Reviewer twqU for commending that our “**methods are reliable**”, our “**dataset represents a valid addition to the LLM community**”, our experiments are “**sound**”, and results “**highlight an open challenge**”. We are glad they found our paper “**primarily well-written**”, echoing Reviewer fmAM.
>
> **Suggestion 1 (S1): “mention how failure could impact LLM real-world applications”**
>
> LLMs’ ability to detect and handle knowledge conflicts is crucial to ensure robustness and guard against misinformation. Suppose a model is given this context, like DS-type rulebreakers in our dataset:``“The patient was informed that the treatment for their illness is available either in Berlin or somewhere in Germany. However, as it turns out, the treatment is not available anywhere in Germany.”``
>
> Models rigidly applying logic would incorrectly conclude that “the treatment is available in Berlin”, contradicting the fact in context. Instead, an ideal model should recognize that, since Berlin is in Germany, the treatment is not available in Berlin either.
>
> To demonstrate, we add the **bold** phrases to premises in our prompts ("**Suppose we are told that** Anne is either in Stockholm or somewhere in Sweden. **However, as a matter of fact,** Anne is not in Sweden"), but found that models' performances on RULEBREAKERS remain mediocre (paired accuracy ranging 0.29 to 0.69). Due to space, we will add and discuss these results in the final version.
>
> **S2/Weakness 1 (W1): “Some findings not clearly presented: [Section 5.2, Section 6, Fig 5]”**
>
> We will replace Fig 5 with a table and revise Section 5.2 to improve clarity. Specifically, we will make clear that we use a model’s output probabilities as a measure of its confidence. We will also add to Section 6 the clarification we set out in our Q2 response to Reviewer QkHy.
>
> **W2: “selected only small/medium-sized LLMs”/Question 1 (Q1): “what would authors expect evaluating larger models?”**
>
> We included a 70B model and the large model GPT-4o. We do not observe a correlation between model size and performance on RULEBREAKERS: GPT-4o underperforms most models, and Llama-3-70B-Instruct underperforms its 8B variant.
> Thus, we consider that our results with selected model sizes **already make a strong point regarding limitations of current LLMs, including state-of-the-art GPT-4o**.
>
> **Q2: “Did authors consider impact of temperature when prompting?”/S3: “add experiments [re] temperature”**
>
> In our main experiment, we run only one forward pass to extract the most probable token without randomly sampling, so temperature is not used.
>
> In Appendix G, we prompt LLMs to generate conclusions by greedy decoding. Our results already make a strong case, but we follow the Reviewer’s suggestion and test a subset of models. As Table 1 shows, random sampling with temperature yielded mixed effects for different models but did not alter our findings. We will add these results in the final version.
>
> | **Temperature** | **Phi-3-mini-128k-Instruct** | **Llama-3-8B-Instruct** | **Mistral-7B-Instruct-v0.3** |
> |---|---|---|---|
> | Not set - greedy decoding (baseline) | 1.23 | **32.44** | **19.50** |
> | 0.1 | 1.47 | 32.34 | 19.26 |
> | 0.5 | 4.24 | 31.76 | 18.68 |
> | 1.0 | **7.59** | 28.51 | 17.12 |
> Table 1. % of correct paired responses
>
> **Q3: “performance of same LLM varies across rulebreaker types...no correlation between performance of different models on different types...any insight on why?”/S4: “add more insights on why these behaviours emerge”**
>
> We appreciate and agree with these observations. We expect models may differ in specific traits: some may be better at reasoning with geographical entities; others worse at recognizing some forms of negated premises. However, the fact that performance varies unpredictably shows that models do not handle these reasoning problems robustly or consistently. Rather, they may be affected by biases or quirks in the training data.
>
> **Q4: “additional experiments implementing hybrid LLM approaches?”/W3: “do not check how these hybrid models perform”**
>
> Our study aimed to expose a blind spot even in general-purpose models. We add a model specifically fine-tuned for logical reasoning (see our Q3 response to Reviewer QkHy), which enhances our findings and reflects our motivation for introducing “rulebreakers”.
>
> Hybrid systems like LogicLM [1] use LLMs to parse natural language statements into logical form. However, **each rulebreaker and non-rulebreaker in a pair share the same surface form and are both valid in propositional logic**. Thus, even if the parsing is correct, the system would theoretically accept the conclusion in both rulebreakers and non-rulebreakers, hence scoring 0 in paired accuracy. We will add this discussion to the final version.
>
> **S5: “better representation of Fig 6”**
>
> Per the Reviewer's suggestion, we will replace Fig 6 with a table of mean familiarity values (and std) for clarity.
>
> [1] Pan et al. https://aclanthology.org/2023.findings-emnlp.248

---

> > ### Comment · Reviewer_twqU · 2025-04-04
> >
> > I thank the authors for their rebuttal. The authors addressed a few of my original doubts, especially the ones concerning the temperature parameter and the effect of model size on the obtained results. However, some other questions and doubts remain open even after the rebuttal. More in detail, the authors seemed to avoid (or at least circumvent) the question concerning the comparison with hybrid models. The authors briefly mention that such models would, in theory, score 0 paired accuracy. However, the authors focused exclusively on LogicLM, which is just one of the different hybrid approaches available. Moreover, such a claim is not supported by empirical evidence (possibly given the short time available for the rebuttal). Similarly, the answer to Q3 is a bit dry, and I'd suggest the authors expand on this explanation. While I agree that biases may arise depending on the training data used, they can not justify the low performance of LLMs on rule-breakers. Intuitively, there should be a deeper reason behind such a brittle performance, possibly related to the inherent regressive nature of LLMs. Could the authors elaborate more on this topic? Finally, I had a look at the other reviewers' insights and the related authors' responses. In particular, I agree with reviewer 8kkH that the relationship between the current submission and the findings of [1] and [2] is not well discussed in the paper. I would not go as far as reviewer 8kkH to say that "the main finding of this paper seems to have already been well-studied by previous works", but I'd suggest the authors to improve the presentation of the paper to better highlight the differences between RULEBREAKERS and [1] and [2].
> >
> > Overall, I think that the rebuttal submitted by the authors covered some of my doubts, while leaving also some open questions. However, such a lack is not enough to justify lowering my review score. Neither the few filled gaps are enough to justify increasing my original review score (which was already positive). Therefore, I will keep my original score.
> >
> > [1]. Reasoning or Reciting? Exploring the Capabilities and Limitations of Language Models Through Counterfactual Tasks
> >
> > [2]. Language models show human-like content effects on reasoning tasks

---

> > > ### Author Response · Authors · 2025-04-08
> > >
> > > We thank the Reviewer for their helpful reply.
> > >
> > > _**“comparison with hybrid models”**_
> > >
> > > To clarify, we consider our theoretical argument to apply generally to any models or systems that are explicitly guided or constrained to reason according to formal logic. However, we are happy to validate this by empirically testing LogicLM and also Symbolic CoT [1] and will include the results in the final version due to time. We welcome additional specific suggestions, though we kindly ask the Reviewer to take into account the prohibitive cost associated with pre-training such models.
> > >
> > >
> > > _**Further response to Q3**_
> > >
> > > As we are the first to identify this systematic phenomenon, we agree that this opens up a rich area for further investigation and analysis in future work.
> > >
> > > For example, one of the patterns we noted in **lines 318-324** is that models generally perform better on the DS subset of RULEBREAKERS compared to the MT subset.
> > >
> > > A possible explanation for this performance difference could be the structural similarity between the natural language premises in different rulebreaker types and their apparent logical forms. Intuitively, for a model to recognize that a logical rule can be applied, it needs to recognize where segments of the natural language statements are negations of one another, e.g. abstracting these segments into **B** and **not B**. This mapping is more obvious and straightforward in MT cases than in DS ones.
> > >
> > > For example, in MT (country, city) cases, the **bolded** and _italicized_ parts form a clear negation pair:
> > >
> > > "If Anne is in Sweden, then **she is not in Stockholm**. _Anne is in Stockholm_." (Logical form: If A, then **not B**. _B_.)
> > >
> > > By contrast, the negation is less obvious in DS (country, city) cases:
> > >
> > > "**Anne is** either in Stockholm or **somewhere in Sweden**. _Anne is not in Sweden_." (A or **B**. _Not B_.)
> > >
> > > We conjecture that this may make it more difficult for models to map DS premises and conclusions to their logical forms, compared to MT ones. As a result, models may be less likely to rigidly apply logical rules, hence avoiding incorrect conclusions in DS rulebreaker cases. Nevertheless, as with other factors such as entity familiarity, the precise impact of this “structural-similarity effect” on model behavior may vary across models, potentially contributing to the performance variation we observe. We will include this expanded discussion in the final version of our paper.
> > >
> > > _**“relationship between the current submission and the findings of [2] and [3]”**_
> > >
> > > We thank the Reviewer for their suggestion. As we explained in our response to Reviewer 8kkH, there appears to be a misunderstanding regarding the relationship between our submission and the findings of [2] and [3]. We explicitly discussed and distinguished [2] from our work in **Appendix A (lines 829-833)**; and similarly addressed Saparov and He (2023) [4], which predates the relevant methodology and findings in [3], in **lines 846-852**. Nonetheless, to ensure full clarity, we will include a more in-depth discussion in the final version of our paper, along with the comparison table we already provided in our response to Reviewer 8kkH.
> > >
> > > [1] Xu et al. 2024. https://aclanthology.org/2024.acl-long.720/
> > >
> > > [2] Reasoning or Reciting?
> > >
> > > [3] Language models show human-like content effects on reasoning tasks
> > >
> > > [4] Saparov and He. https://openreview.net/pdf?id=qFVVBzXxR2V

---

### Decision · Program_Chairs · 2025-05-01

**Decision:**

Accept (poster)

**Comment:**

This paper contributes a dataset designed to assess LLMs' ability to reason using common sense and factual knowledge rather than strictly following formal logic. Most of the reviewers agree that the work is of high quality, and the dataset + evaluation is well executed, though there is a large score discrepancy. This is because the focus of the work has strong similarities to prior work by Lampinen et al. and by Wu et al., which test a different but closely related hypothesis. The author do discuss this prior work and highlight differences, but the reviewers can not agree whether this is reason to recommend rejection or not. After reading the reviews, I am inclined to agree with the summary of Reviewer fmAM who highlights the merits of this work even though there are similarities:

> the authors have written very clearly how they construct their dataset, which is the main focus of their work. They also use a templated dataset, which makes it harder for language models to just learn the answers and also more extensible. Their evaluation shows the benefits of their dataset. The other papers in comparison rely on small datasets and/or manually written datapoints. I therefore think their paper deserves a fair chance and disagree with the negative view of reviewer 8kkH. In my opinion, the main focus of the article is the dataset, which the authors describe in detail and their description is well-written and easy to follow. So even if the authors can not fully explain the trends, as reviewer QkHy remarks, seen in their analysis, the evaluation shows the necessity for such a dataset.

Considering the above, together with the broad agreement (except Reviewer QkHy) about the quality of this work (ignoring the prior work issue) I will mark this work as leaning accept.